# SHIELD: Suppressing Hallucinations In lvlm Encoders via bias and vuLnerability Defense

**Yiyang Huang**[1] **Liang Shi**[1] **Yitian Zhang**[1] **Yi Xu**[1] **Yun Fu**[1,2]

[1]Department of Electrical and Computer Engineering, Northeastern University

[2]Khoury College of Computer Science, Northeastern University

{huang.yiyan,shi.lia,zhang.yitian,xu.yi,y.fu}@northeastern.edu

## Abstract

Large Vision-Language Models (LVLMs) excel in diverse cross-modal tasks. However, object hallucination, where models produce plausible but inaccurate object descriptions, remains a significant challenge. In contrast to previous work focusing on LLM components, this paper is the first to trace LVLM hallucinations to visual encoders and identifies three key issues: statistical bias, inherent bias, and vulnerability. To address these challenges, we propose SHIELD, a training-free framework that mitigates hallucinations through three strategies: re-weighting visual tokens to reduce statistical bias, introducing noise-derived tokens to counter inherent bias, and applying adversarial attacks with contrastive decoding to address vulnerability. Experiments demonstrate that SHIELD effectively mitigates object hallucinations across diverse benchmarks and LVLM families. Moreover, SHIELD achieves strong performance on the general LVLM benchmark, highlighting its broad applicability. *Code is available at https://github.com/hukcc/SHIELD.*

## 1 Introduction

Large Vision-Language Models (LVLMs) (Bai et al., 2023b; Liu et al., 2023b; Dai et al., 2023) combine visual and textual information and have advanced significantly in cross-modal tasks. Despite these advances, they suffer from object hallucination, generating object descriptions that may appear reasonable but misrepresent the image, either by misidentifying attributes (e.g., color, quantity, position) or by introducing non-existent objects. This issue poses reliability and safety risks in domains such as healthcare (Hu et al., 2023; Wang et al., 2023b), autonomous systems (Chen et al., 2024a; Wu et al., 2023), and robotics (Mai et al., 2023; Liu et al., 2023a).

Various approaches have been proposed to mitigate object hallucinations. Early efforts, such as fine-grained modality alignment (Biten et al., 2022) and data augmentation to reduce co-occurrence bias (Kim et al., 2023; Rohrbach et al., 2018), were designed for small-scale VLMs but fail to generalize to LVLMs (Kaplan et al., 2020; Wei et al., 2022). More recent research falls into two categories: training-required and training-free methods. Training-required methods, including preference optimization (Ouali et al., 2024), post-hoc revisers (Zhou et al., 2024), and RLHF (Sun et al., 2024a),

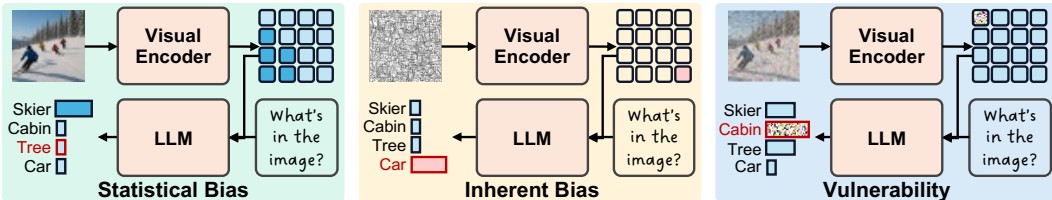

Figure 1: Key issues underlying object hallucinations in LVLMs. Statistical bias: the visual encoder overemphasizes frequent visual patterns, distorting fine-grained perception. Inherent bias: the encoder produces erroneous representations of dominant objects in the pretraining data, regardless of input. Vulnerability: the encoder is sensitive to minor perturbations, yielding unreliable features.

improve factual consistency but demand substantial human and computational resources. In contrast, training-free methods, such as contrasting outputs from distorted inputs (Leng et al., 2024) or applying over-trust penalties during decoding (Huang et al., 2024), offer a more efficient alternative. However, these approaches primarily focus on the LLM component, leaving the role of visual encoders underexplored.

This paper is the first to trace LVLM hallucinations to visual encoders, filling this gap by identifying three key issues: statistical bias, inherent bias, and vulnerability, as illustrated in Figure 1. Despite large-scale pretraining, these encoders remain affected by imbalanced distributions of visual concepts in the pretraining data, resulting in statistical and inherent biases. Statistical bias leads the visual encoder to overemphasize tokens related to frequent visual patterns, thereby distorting the perception of details. Inherent bias leads the visual encoder to produce representations of dominant objects in the pretraining data, regardless of the input, even when it is meaningless. Furthermore, vulnerability, arising from insufficient robustness to noise and perturbations during pretraining, leads the encoder to produce inaccurate visual representations even with small perturbations.

To address bias and vulnerability in visual encoders that hinder feature extraction and amplify hallucinations in LVLMs, we propose SHIELD, a training-free method combining token re-weighting, token subtraction, and contrastive decoding. Specifically, token re-weighting alleviates statistical bias by distributing attention across more tokens relevant to the ground-truth objects, thus avoiding fine-grained distortion from overemphasized tokens. In parallel, token subtraction mitigates inherent bias by estimating erroneous representations related to dominant objects in pretraining data via noise input and eliminating them through token-level subtraction. To address vulnerability, contrastive decoding exposes hallucinations with a perturbed image and suppresses them by contrasting with outputs from a natural image.

Experiments demonstrate that SHIELD consistently improves performance on object hallucination benchmarks, including CHAIR (Rohrbach et al., 2018), POPE (Li et al., 2023), the hallucination subset of MME (Fu et al., 2023), and GPT-4o-aided evaluations on LLaVA-Bench (Liu et al., 2023b). Moreover, these improvements are observed across diverse LVLM families, such as LLaVA (Liu et al., 2023b), InstructBLIP (Dai et al., 2023), and Qwen-VL (Bai et al., 2023b). Beyond object hallucination mitigation, SHIELD also enhances general perception capabilities, as evidenced by improvements on the full MME benchmark (Fu et al., 2023), highlighting its broader applicability.

Our contributions are summarized as follows:

- We analyze the role of visual encoders in contributing to object hallucinations in LVLMs, focusing on statistical bias, inherent bias, and vulnerability.
- We propose SHIELD, a training-free method that mitigates object hallucinations by reducing statistical bias via token re-weighting, alleviating inherent bias using token subtraction, and addressing vulnerability through contrastive decoding.
- Comprehensive experiments validate SHIELD's effectiveness in mitigating object hallucinations across diverse benchmarks and LVLM families. Moreover, its strong performance on the general LVLM benchmark highlights its broad applicability.

## 2 RELATED WORK

### 2.1 LARGE VISION LANGUAGE MODELS (LVLMS)

Recent advances in large-scale foundation models and multimodal learning have accelerated the development of Large Vision-Language Models (LVLMs). By combining Large Language Models (LLMs) (Bai et al., 2023a; Brown et al., 2020; Chiang et al., 2023; Chowdhery et al., 2023; Gilardi et al., 2023; Raffel et al., 2020; Taori et al., 2023; Tay et al., 2023; Touvron et al., 2023) with cross-modal frameworks such as CLIP (Radford et al., 2021) and BLIP (Li et al., 2022), LVLMs integrate visual and textual information for more comprehensive understanding. Nevertheless, LVLMs across different architectures, including LLaVA-1.5 (Liu et al., 2024), InstructBLIP (Dai et al., 2023), and Qwen-VL (Bai et al., 2023b), still suffer from hallucinations, particularly in fine-grained object recognition and challenging visual grounding. Such errors, often involving non-existent objects or misidentified attributes, remain a key challenge for reliability in real-world applications.

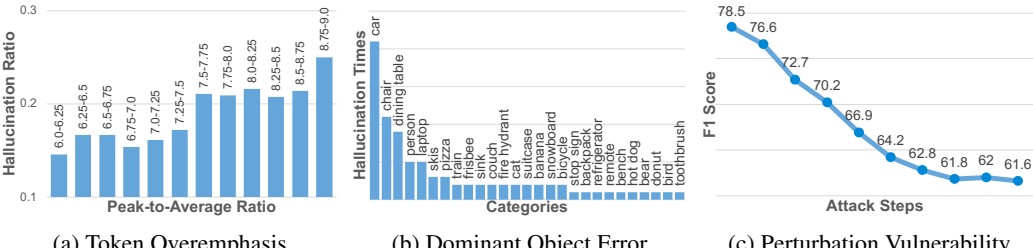

Figure 2: Statistics show that hallucinations stem from bias and vulnerability. (a) The X-axis shows the peak-to-average L2 norm ratio of visual tokens, measuring the deviation of the highest-norm token from the average, and the Y-axis shows the proportion of hallucinating samples at each level. Stronger overemphasis leads to higher hallucination rates. (b) The X-axis lists objects, and the Y-axis shows hallucination occurrences under meaningless inputs. Dominant objects are more likely to be falsely perceived as present. (c) The X-axis denotes the number of attack steps, and the Y-axis shows the F1 score. Even small perturbations increase hallucinations and degrade performance.

## 2.2 OBJECT HALLUCINATION IN LVLMS

Approaches to mitigating object hallucinations in LVLMs can be grouped into training-required and training-free methods. Training-required methods reduce hallucinations by optimizing model parameters or training auxiliary modules. Prominent methods include CLIP-DPO (Ouali et al., 2024), leveraging CLIP-based similarity ranking for preference optimization; LURE (Zhou et al., 2024), using post-hoc revisers to align text with visual input; and LLaVA-RLHF (Sun et al., 2024a), incorporating human feedback through reinforcement learning. Training-free methods improve decoding without modifying model. Representative approaches include Visual Contrastive Decoding (VCD) (Leng et al., 2024), which contrasts outputs from natural and blur inputs to mitigate hallucinations, OPERA (Huang et al., 2024), which reduces overconfidence through penalty mechanisms and token-level adjustments, HALC (Chen et al., 2024b), which employs adaptive focal-contrast decoding to provide token-wise visual grounding during inference, MARINE (Zhao et al., 2025a), which incorporates image-grounded guidance from external vision models, and VTI (Liu et al., 2025), which steers latent representations to stabilize vision features at test time. While effective, these methods rarely address the bias and vulnerability of visual encoders, which this work aims to address.

## 3 METHOD

### 3.1 HALLUCINATIONS STEM FROM VISUAL ENCODER

Accurate visual feature extraction is crucial for LVLMs to generate reliable outputs. However, bias and vulnerability in visual encoders distort features, intensifying object hallucinations. This section delves into these challenges.

#### 3.1.1 STATISTICAL AND INHERENT BIAS IN VISUAL ENCODER

Most LVLMs adopt visual encoders derived from pretrained CLIP models. Although these encoders benefit from large-scale pretraining, they are influenced by the imbalanced distribution of visual concepts in the pretraining data. Specifically, certain visual concepts appear far more frequently than others, while rare or context-dependent elements are severely underrepresented (Parashar et al., 2024). As a result, the model develops a strong inductive bias toward frequent patterns and dominant objects, giving rise to both statistical and inherent bias.

Statistical bias denotes the visual encoder's over-reliance on frequent visual patterns in the pretraining data, causing overemphasis on the corresponding tokens with disproportionately high L2 activation values (Darcet et al., 2024). This overemphasis distorts the downstream LLM's perception of fine-grained details by directing attention to overweighted tokens (Kang et al., 2025), often resulting in hallucinations. Analysis of LLaVA-1.5's responses and visual tokens on the POPE COCO subset (Figure 2a) shows that the proportion of hallucinated samples grows with stronger token overemphasis, measured by the peak-to-average L2 ratio (the deviation of the highest-norm token from the mean among visual tokens).

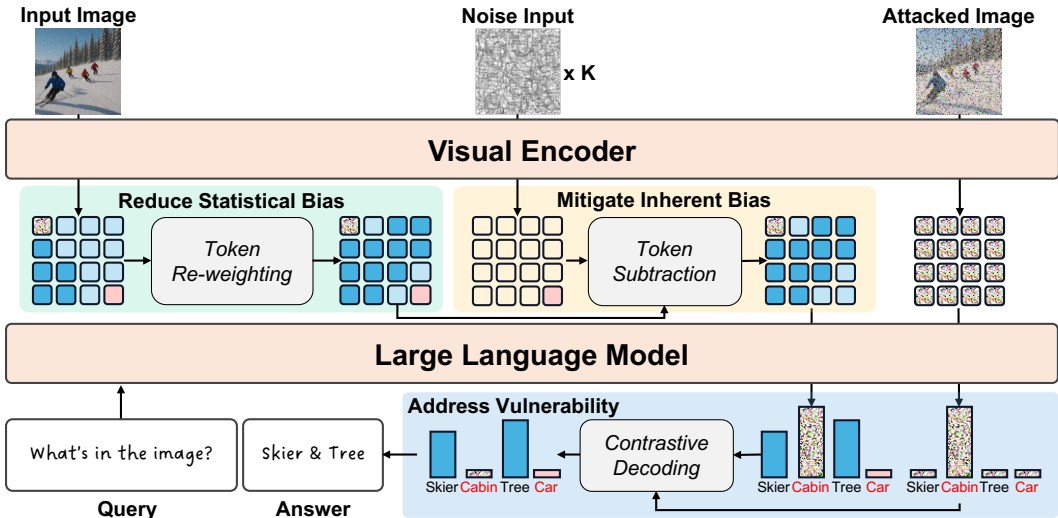

Figure 3: Illustration of the proposed SHIELD framework. Given an input image and a query text, the visual encoder produces tokens affected by statistical bias (overemphasized tokens ▢), inherent bias (erroneous representations ▢), and vulnerability (inaccurate features ▢). SHIELD addresses these issues through three modules: (i) **Token Re-weighting**, which redistributes attention to more ground-truth-object relevant tokens to alleviate overemphasis (▢); (ii) **Token Subtraction**, which estimates and removes erroneous representations (▢) via noise-derived tokens; and (iii) **Contrastive Decoding**, which exposes inaccurate features (▢) using attacked images and suppresses corresponding outputs by contrasting them with those from the natural image.

Inherent bias is the visual encoder's overdependence on dominant objects in the pretraining data, leading it to generate erroneous representations of these objects regardless of the input, even when meaningless. As shown in Figure 2b, analysis of LLaVA-1.5's responses to the POPE COCO random split questions with meaningless images (random noise) as input shows frequent hallucinations of dominant objects such as cars, chairs, and tables, defined as cases where the model incorrectly predicts the presence of queried objects.

### 3.1.2 VULNERABILITY IN VISUAL ENCODER

The vulnerability of visual encoders is another key factor contributing to object hallucinations. It arises from their limited robustness to noise and subtle perturbations (Mao et al., 2023), making them susceptible to constructing inaccurate visual representations under such disturbances. As shown in Figure 2c, the performance of LLaVA-1.5 drops sharply on the POPE COCO subset even with a few attack steps, demonstrating that minor perturbations can exploit this weakness and yield unreliable features.

### 3.2 SHIELD: SUPPRESSING HALLUCINATIONS IN LVLM ENCODERS VIA BIAS AND VULNERABILITY DEFENSE

Building on these observations, we propose SHIELD, a training-free method to mitigate object hallucinations by addressing statistical bias, inherent bias, and vulnerability in visual encoders, as illustrated in Figure 3. SHIELD integrates three strategies. Token re-weighting distributes attention across more tokens relevant to ground-truth objects, thereby reducing fine-grained distortion from overemphasized tokens and alleviating statistical bias. Token subtraction estimates erroneous representations of dominant objects in the pretraining data using noise input and removes them through token-level subtraction, thus mitigating inherent bias. Finally, contrastive decoding applies perturbations to the input image to expose hallucinations and suppresses them by contrasting outputs with those from the natural image, countering vulnerability.

Figure 4: Mitigating Statistical Bias. Visual tokens are re-weighted via a similarity matrix between visual tokens and naive caption tokens, emphasizing more ground-truth-object relevant tokens and reducing overemphasis.

Figure 5: Reducing Inherent Bias. $K$ noise inputs are used to estimate erroneous representations of dominant objects in the pretraining data, which are then removed from visual tokens via feature subtraction.

### 3.2.1 FORMULATION OF LVLM INFERENCE

A LVLM's inference can be described in three stages. First, the visual encoder $E(\cdot)$ extracts $N$ visual tokens from the raw image $\mathbf{v}$:

$$\mathbf{x}^v = x_0, x_1, \ldots, x_{N-1} = E(\mathbf{v}). \tag{1}$$

Next, given $\mathbf{x}^v$ and the query text $\mathbf{t}$, the LLM computes the output logits for token $y_i$ at step $i$, conditioned on the preceding sequence $y_{<i}$:

$$\text{logit}(y_i \mid \mathbf{x}^v, \mathbf{t}, y_{<i}) = \text{LLM}(\mathbf{x}^v, \mathbf{t}, y_{<i}). \tag{2}$$

Finally, the logits are transformed into a probability distribution over the vocabulary via softmax, from which the next token is selected according to the decoding strategy:

$$p(y_i \mid \mathbf{x}^v, \mathbf{t}, y_{<i}) = \text{softmax}\left[\text{logit}(y_i \mid \mathbf{x}^v, \mathbf{t}, y_{<i})\right]. \tag{3}$$

Autoregressive repetition of the second and third stages produces the final textual output.

### 3.2.2 MITIGATING STATISTICAL BIAS

As discussed in Section 3.1.1, statistical bias causes the visual encoder to overemphasize tokens associated with frequent visual patterns, distorting fine-grained perception. To address this, token re-weighting is applied based on the similarity between visual tokens and naive caption tokens, encouraging the model to attend to more tokens relevant to ground-truth objects.

As shown in Figure 4, token re-weighting begins with generating a naive caption $\mathbf{c}^{\text{naive}}$ using the vanilla LVLM for the given image $\mathbf{v}$:

$$\mathbf{c}^{\text{naive}} = \text{VanillaLVLM}\left(\mathbf{v}, \text{``Please describe this image''}\right). \tag{4}$$

The CLIP text encoder $E_t(\cdot)$ (paired with $E(\cdot)$ during CLIP pretraining) then encodes the caption into $P$ tokens:

$$\mathbf{c} = \{c_0, c_1, \ldots, c_{P-1}\} = E_t(\mathbf{c}^{\text{naive}}). \tag{5}$$

Given the caption tokens $\mathbf{c}$ and the visual tokens $\mathbf{x}^v$, a similarity matrix $\mathbf{M} \in \mathbb{R}^{N \times P}$ is computed:

$$\mathbf{M} = \frac{\mathbf{x}^v \mathbf{c}^\top}{\|\mathbf{x}^v\|_2 \cdot \|\mathbf{c}\|_2}. \tag{6}$$

From $\mathbf{M}$, weights $\mathbf{W}^v$ are obtained by taking the maximum along the caption dimension and normalizing to $[0, 1]$:

$$\mathbf{W}^v = \text{norm}(\max_j \mathbf{M}_{i,j}), \quad \mathbf{W}^v \in \mathbb{R}^N. \tag{7}$$

Finally, the weights are applied via residual addition ($\odot$: element-wise multiplication) to emphasize visual tokens corresponding to captioned objects, yielding statistical-bias-corrected tokens $\mathbf{x}^{v\prime}$:

$$\mathbf{x}^{v\prime} = \mathbf{x}^v + \mathbf{x}^v \odot \mathbf{W}^v. \tag{8}$$

Although the naive caption $\mathbf{c}^{\text{naive}}$ may introduce hallucinations, they do not affect re-weighting, as hallucinated objects fail to match any visual tokens with high similarity during similarity matrix $\mathbf{M}$ computation. Thus, token re-weighting remains focused on ground-truth objects.

### 3.2.3 REDUCING INHERENT BIAS

As in Section 3.1.1, inherent bias leads the visual encoder to produce erroneous representations of dominant objects in the pretraining data, regardless of the input. To counter this, token subtraction introduces noise inputs to estimate such erroneous features and removes them from the visual tokens.

As shown in Figure 5, $K$ random noise inputs $\mathbf{n}_i$ (with the same size as the image) are passed through the visual encoder. The resulting tokens are averaged to estimate erroneous representations, which are then subtracted from the statistical-bias-corrected tokens $\mathbf{x}^{v\prime}$, yielding bias-reduced tokens:

$$\mathbf{x}^{v\prime\prime} = \mathbf{x}^{v\prime} - \frac{1}{K}\sum_{i=1}^{K} E(\mathbf{n}_i). \tag{9}$$

Since inherent bias depends only on visual encoder parameters, the estimation of erroneous representations from noise inputs can be pre-calculated for each model to improve efficiency.

The bias-reduced tokens $\mathbf{x}^{v\prime\prime}$, together with the query text $\mathbf{t}$, are subsequently fed into the LLM to produce bias-reduced logits:

$$\text{logit}\left(y_i \mid \mathbf{x}^{v\prime\prime}, \mathbf{t}, y_{<i}\right) = \text{LLM}\left(\mathbf{x}^{v\prime\prime}, \mathbf{t}, y_{<i}\right). \tag{10}$$

### 3.2.4 ADDRESS VULNERABILITY

As noted in Section 3.1.2, the visual encoder lacks robustness to subtle perturbations and noise, making it susceptible to inaccurate representations, especially when key pixels are disturbed. To counter this vulnerability, a two-step strategy is adopted: adversarial attack is first applied to reveal objects likely to be hallucinated, followed by contrastive decoding to suppress the probability of generating the corresponding outputs during inference.

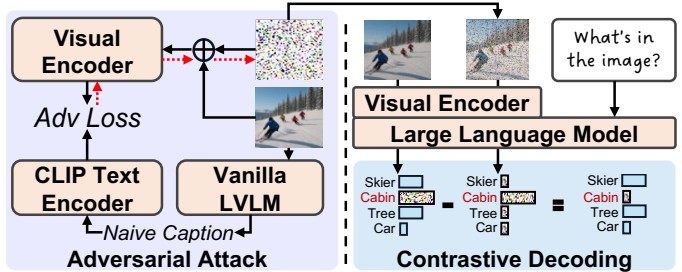

Figure 6: Addressing Vulnerability. An attack tensor constructed from the input image and its naive caption via adversarial learning is applied to reveal objects likely to be hallucinated, followed by contrastive decoding to suppress their generation.

To expose vulnerability-induced hallucinations, an attack tensor is constructed from the input image $\mathbf{v}$ and its naive caption $\mathbf{c}^{\text{naive}}$ (Equation 4) using the visual encoder $E(\cdot)$ and its paired text encoder $E_t(\cdot)$. As illustrated in Figure 6, a learnable perturbation $\delta$ is added to the input image and refined via backpropagation. The adversarial loss is defined as the cosine similarity between the global representation of the perturbed image and that of the naive caption:

$$l_{\text{adv}} = \cos\left(E(\mathbf{v} + \delta),\ E_t(\mathbf{c}^{\text{naive}})\right). \tag{11}$$

The final attack tensor $\delta^*$ is obtained by minimizing $l_{\text{adv}}$ via gradient descent with learning rate $l$:

$$\delta^* = \arg\min_{\delta}\ l_{\text{adv}}. \tag{12}$$

During inference, the attack tensor $\delta^*$ is added to the input image to produce vulnerability-induced inaccurate visual representations:

$$\overline{\mathbf{x}}^v = \{\bar{x}_0, \bar{x}_1, \ldots, \bar{x}_{N-1}\} = E(\mathbf{v} + \delta^*). \tag{13}$$

These inaccurate representations are then used to produce adversarial logits:

$$\text{logit}\left(y_i \mid \overline{\mathbf{x}}^v, \mathbf{t}, y_{<i}\right) = \text{LLM}\left(\overline{\mathbf{x}}^v, \mathbf{t}, y_{<i}\right). \tag{14}$$

To suppress hallucinations revealed by the attack tensor, contrastive decoding is applied. At each decoding step $i$, SHIELD contrasts the bias-reduced logits with the adversarial logits to adjust the output probability distribution, where $\alpha$ controls the impact of contrastive decoding:

$$p_{shield}(y_i) = \text{softmax}\Big[(1+\alpha)\,\text{logit}(y_i \mid \mathbf{x}^{v\prime\prime}, \mathbf{t}, y_{<i}) - \alpha\,\text{logit}(y_i \mid \overline{\mathbf{x}}^v, \mathbf{t}, y_{<i})\Big], \tag{15}$$

Table 1: CHAIR Hallucination Evaluation

| Method | LLaVA-1.5 | | InstructBLIP | | Qwen-VL | |
|---|---|---|---|---|---|---|
| | $C_S\downarrow$ | $C_I\downarrow$ | $C_S\downarrow$ | $C_I\downarrow$ | $C_S\downarrow$ | $C_I\downarrow$ |
| Vanilla | 48.8 | 14.2 | 54.6 | 24.8 | 49.2 | 13.1 |
| VCD | 46.8 | 13.2 | 44.0 | 13.6 | 46.4 | 11.9 |
| OPERA | 44.6 | 12.8 | 46.4 | 14.2 | 34.6 | 9.5 |
| Ours | **36.6** | **10.3** | **40.4** | **10.9** | **28.9** | **9.2** |

Table 2: GPT4o-aid Hallucination Evaluation

| Method | LLaVA-1.5 | | InstructBLIP | | Qwen-VL | |
|---|---|---|---|---|---|---|
| | $C\uparrow$ | $D\uparrow$ | $C\uparrow$ | $D\uparrow$ | $C\uparrow$ | $D\uparrow$ |
| Vanilla | 4.9 | 5.0 | 4.2 | 4.2 | 6.2 | 4.6 |
| VCD | 5.5 | 5.5 | 5.1 | **5.5** | 6.5 | 5.7 |
| OPERA | 5.6 | 6.0 | 5.3 | 5.2 | 6.5 | 5.6 |
| Ours | **6.2** | **6.1** | **5.6** | 5.3 | **6.9** | **5.8** |

Following (Leng et al., 2024), an adaptive plausibility constraint is introduced to avoid implausible outputs. Only tokens with probabilities no smaller than a fraction $\beta$ of the maximum are retained:

$$\nu_{\text{token}}(y_i) = \{\, y_i \in \nu : p(y_i) \geq \beta \max_\omega p(\omega) \,\}, \tag{16}$$

where $\nu$ is the vocabulary and $\nu_{\text{token}}(y_i)$ is the valid subset at step $i$. The threshold $\beta$ determines truncation aggressiveness. For tokens not in $\nu_{\text{token}}(y_i)$, probabilities are set to zero.

## 4 EXPERIMENTS

### 4.1 IMPLEMENTATION DETAILS

To evaluate the effectiveness of SHIELD in mitigating hallucinations, three representative LVLMs were selected: LLaVA-1.5 (Liu et al., 2024), InstructBLIP (Dai et al., 2023), and Qwen-VL (Bai et al., 2023b). SHIELD was compared against the corresponding vanilla LVLMs and two recent training-free methods, VCD (Leng et al., 2024) and OPERA (Huang et al., 2024). Following their original setups, vanilla LVLMs and VCD adopted sampling-based decoding, while OPERA employed beam search decoding with a penalty term on logits to reduce overconfidence. For SHIELD, sampling-based decoding was used, drawing from the modified post-softmax distribution. Unless otherwise specified, $\alpha = 2$, $\beta = 0.35$, $K = 32$, and $l = 0.02$ were applied across all LVLMs, where $\alpha$ controls the strength of contrastive decoding, $\beta$ sets the truncation threshold in the plausibility constraint, $K$ denotes the number of noise inputs for estimating inherent bias, and $l$ is the learning rate for optimizing the attack tensor. Hyper-parameters ablation are provided in the Appendix C. All experiments were conducted on a single RTX A6000 GPU.

### 4.2 QUANTITATIVE RESULTS

This section evaluates the effectiveness of SHIELD in mitigating hallucinations for both detailed descriptions and simplified VQA answers.

**CHAIR Evaluation.** The CHAIR metric (Rohrbach et al., 2018) measures object hallucination in image captioning by calculating the proportion of objects mentioned in captions that are not present in the ground-truth label set. It consists of two dimensions: a sentence-level score $C_S$ and an instance-level score $C_I$, which are defined as:

$$C_S = \frac{|\text{sentences with hallucinated objects}|}{|\text{all sentences}|}, \quad C_I = \frac{|\text{hallucinated objects}|}{|\text{all objects mentioned}|}.$$

Following (Huang et al., 2024), evaluation is performed on 500 randomly selected images from the COCO 2014 validation set (Lin et al., 2014), using the prompt *"Please describe this image in detail."* To ensure fairness, generated captions are truncated to a maximum of 512 tokens.

As shown in Table 1, SHIELD consistently outperforms all previous training-free methods on both $C_S$ and $C_I$, achieving up to 18% improvement over the second-best method, OPERA, on LLaVA-1.5. This performance gain stems from SHIELD's ability to counter biases and vulnerability in the visual encoder, thereby reducing hallucination risk in detailed descriptions.

**GPT-4o Assisted Evaluation.** While CHAIR effectively evaluates object-level hallucinations, it fails to capture errors in attributes, locations, or relations. To complement this, we employ GPT-4o, a strong multi-modal assistant, to assess LVLM outputs on the LLaVA-Bench dataset. GPT-4o scores responses on correctness (C) and detailedness (D) from 0–10, with higher correctness indicating fewer hallucinations. The evaluation explicitly targets objects mentioned but absent from the image, as well as errors in attributes, colors, positions, or relationships. Further details are provided in the Appendix A.1.

Table 3: POPE Hallucination Evaluation on COCO subset

| LVLM | Method | Random | | Popular | | Adversarial | | Average | |
|---|---|---|---|---|---|---|---|---|---|
| | | Accuracy ↑ | F1 ↑ | Accuracy ↑ | F1 ↑ | Accuracy ↑ | F1 ↑ | Accuracy ↑ | F1 ↑ |
| LLaVA-1.5 | Vanilla | 83.2 | 81.3 | 81.8 | 80.0 | 78.9 | 77.5 | 81.3 | 79.6 |
| | VCD | 87.7 | 87.1 | 85.3 | 85.0 | 80.8 | 81.3 | 84.6 | 84.4 |
| | OPERA | 89.1 | 89.0 | 86.0 | 86.3 | 79.1 | 80.9 | 84.7 | 85.4 |
| | Ours | **91.3** | **91.1** | **87.4** | **87.6** | **82.5** | **83.6** | **87.0** | **87.4** |
| InstructBLIP | Vanilla | 80.7 | 80.4 | 78.2 | 78.3 | 75.8 | 76.5 | 78.2 | 78.4 |
| | VCD | 84.5 | 83.6 | 81.4 | 81.0 | 79.5 | 79.5 | 81.8 | 81.3 |
| | OPERA | **89.8** | **89.6** | 83.4 | 84.0 | 80.7 | 81.8 | 84.6 | **85.1** |
| | Ours | 88.2 | 87.6 | **84.6** | **84.3** | **82.2** | **82.4** | **85.0** | 84.8 |
| Qwen-VL | Vanilla | 84.7 | 82.6 | 84.1 | 82.0 | 82.2 | 80.3 | 83.6 | 81.6 |
| | VCD | 88.6 | 87.8 | 87.1 | 86.4 | 84.2 | 83.9 | 86.6 | 86.0 |
| | OPERA | 86.1 | 84.2 | 85.7 | 83.8 | 83.9 | 82.1 | 85.2 | 83.3 |
| | Ours | **89.2** | **88.6** | **87.6** | **87.1** | **84.3** | **84.2** | **87.0** | **86.6** |

Table 4: MME Hallucination Evaluation

| LVLM | Method | Object-level | | Attribute-level | | Total Score ↑ |
|---|---|---|---|---|---|---|
| | | Existence Score ↑ | Count Score ↑ | Position Score ↑ | Color Score ↑ | |
| LLaVA-1.5 | Vanilla | 175.6 | 124.6 | 114.0 | 151.0 | 565.3 |
| | VCD | 184.6 | 138.3 | 128.6 | 153.0 | 604.6 |
| | OPERA | 180.6 | 133.3 | 123.3 | 155.0 | 592.3 |
| | Ours | **195.0** | **141.6** | **148.3** | **183.3** | **668.3** |
| InstructBLIP | Vanilla | 141.0 | 75.3 | 66.6 | 97.3 | 380.3 |
| | VCD | 168.3 | **92.3** | 64.0 | 123.0 | 447.6 |
| | OPERA | 156.0 | 78.3 | 55.0 | 95.0 | 384.3 |
| | Ours | **170.0** | 75.0 | **88.3** | **128.3** | **461.6** |
| Qwen-VL | Vanilla | 155.0 | 127.6 | 131.6 | 173.0 | 587.3 |
| | VCD | 156.0 | 131.0 | 128.0 | 181.6 | 596.6 |
| | OPERA | 165.0 | 145.0 | **133.3** | 180.0 | 623.3 |
| | Ours | **180.0** | **170.0** | 128.3 | **190.0** | **668.3** |

As shown in Table 2, SHIELD achieves substantial gains in correctness, confirming its effectiveness in mitigating hallucinations. In contrast, improvements in detailedness are modest, since the method primarily addresses bias and vulnerability in the visual encoder rather than enhancing fine-grained descriptive coverage.

**POPE Evaluation.** Similar to CHAIR, POPE (Li et al., 2023) evaluates existence-level hallucinations in LVLMs. It adopts a VQA-style format (e.g., "Is there a {object} in the image?") to test whether models correctly associate images with specific objects. POPE includes three splits: "random" for random objects, "popular" for frequent objects, and "adversarial" for objects semantically related to those in the image. The evaluation is conducted on three subsets: COCO, A-OKVQA, and GQA. Additional results are provided in Appendix B.1.

As shown in Table 3, SHIELD outperforms previous training-free methods across most splits of the COCO subset. Although all methods show performance drops from Random to Adversarial, SHIELD more effectively mitigates hallucinations in the challenging Adversarial split, highlighting that biases and vulnerability in visual encoders are major contributors to hallucinations. For InstructBLIP, however, the improvements are limited since its Q-Former module constrains the use of modified visual features, thereby diminishing the benefits of SHIELD.

**MME Hallucination Subset Evaluation.** Although POPE adopts a VQA format effective for evaluating object-existence-level hallucinations, it does not capture attribute-level aspects such as count, position, and color. To address this limitation, the MME hallucination subsets (Fu et al., 2023) provide a more comprehensive benchmark. Following (Yin et al., 2023), we evaluate object-level hallucinations using the existence and count subsets, and attribute-level hallucinations using the position and color subsets. Performance is reported using the combined metrics of accuracy and accuracy+ as defined in the official implementation.

As shown in Table 4, SHIELD achieves consistent improvements across all models, leading to higher total scores. By correcting statistical bias in visual encoders, SHIELD reduces the impact of overemphasized tokens on fine-grained perception, thereby significantly mitigating attribute-level hallucinations.

**Attribute and Relation Level Hallucination Evaluation.** To further evaluate hallucinations at the attribute and relation level, we conduct experiments on the AMBER benchmark (Wang et al., 2023a), comparing against a series of baselines based on LLaVA-1.5 7B. AMBER offers a comprehensive

Table 5: AMBER Hallucination Evaluation

| Method | Generative Task | | | | Discriminative Task | | | | AMBER Score |
|---|---|---|---|---|---|---|---|---|---|
| | CHAIR↓ | Cover↑ | Hallucination↓ | Cognition↓ | Accuracy↑ | Precision↑ | Recall↑ | F1↑ | |
| Vanilla | 9.2 | 41.3 | 29.2 | 3.7 | 65.7 | 83.2 | 64.7 | 73.2 | 82.0 |
| VCD | 8.1 | 44.2 | 28.6 | 3.1 | 68.3 | 85.8 | 65.2 | 74.0 | 82.9 |
| OPERA | 8.3 | 43.1 | 31.2 | 2.9 | 76.0 | 79.2 | **83.8** | 81.4 | 86.5 |
| Ours | **6.4** | **46.1** | **25.1** | **1.8** | **78.3** | **89.1** | 76.6 | **82.4** | **88.0** |

Table 6: MME Full Set Evaluation

| Method | Perception↑ | Cognition↑ | Total Score↑ |
|---|---|---|---|
| Vanilla | 1279.2 | 352.9 | 1632.1 |
| VCD | 1363.9 | **353.2** | 1717.1 |
| OPERA | 1413.0 | 304.2 | 1717.2 |
| Ours | **1473.0** | 337.8 | **1810.8** |

Table 7: Module Ablation on CHAIR

| Module | $C_S$↓ | $C_I$↓ |
|---|---|---|
| Vanilla LLaVA-1.5 | 48.8 | 14.2 |
| + adaptive plausibility constraint | 50.2 | 13.8 |
| + address vulnerability (Ours) | 46.4 | 12.8 |
| + mitigate statistical bias (Ours) | 40.4 | 11.0 |
| + reduce inherent bias (Ours) | **36.6** | **10.3** |

suite of generative and discriminative metrics, explicitly covering existence, attribute, and relation hallucinations, making it a strong benchmark for assessing multimodal hallucination across diverse tasks. As shown in Table 5, our method achieves the highest Amber Score (88.0) and consistently outperforms all baselines across most metrics, demonstrating effective mitigation of hallucinations.

**LVLM General Evaluation.** To evaluate the overall performance of SHIELD-enhanced LVLMs, we conduct experiments on the full MME benchmark (Fu et al., 2023) using the LLaVA-1.5 7B model. The benchmark covers ten perception-related subtasks (including four hallucination-related) and four cognition-oriented ones. Performance is reported using both accuracy and accuracy+ as defined in the official implementation.

As shown in Table 6 and Figure 7, SHIELD not only improves hallucination-related performance but also yields notable gains in perception tasks such as OCR and Posters, thereby enhancing the overall capability of the model. Further details are provided in the Appendix B.2.

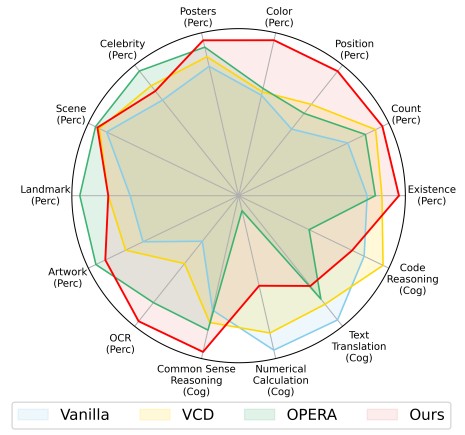

Figure 7: Evaluation on the full MME. Larger radar indicate better performance.

## 4.3 ABLATION STUDY

**Module Ablation.** To assess the effectiveness of SHIELD, we performed ablation studies on each module using the CHAIR benchmark with the LLaVA-1.5 7B model. The adaptive plausibility constraint, a key element of contrastive decoding, was also ablated to evaluate its role in mitigating hallucinations. As shown in Table 7, all modules contribute notably to reducing hallucinations.

Although integral to contrastive decoding, the adaptive plausibility constraint alone was less effective, indicating that filtering low-probability candidates cannot fully suppress hallucinations. In contrast, each SHIELD module individually reduced hallucination frequency, with their full combination achieving the greatest improvement. Notably, after addressing vulnerability, adding the statistical bias mitigation module yielded a further 13% reduction, highlighting statistical bias as a major source of hallucinations, particularly in longer descriptions.

**Visualization.** Figure 8 illustrates SHIELD's effectiveness in mitigating biases and reducing vulnerability in visual encoders. Figure 8a demonstrates that re-weighting visual tokens alleviates statistical bias by distributing attention across more object-relevant tokens and reducing overemphasis on specific ones, thereby improving fine-grained perception. Figure 8b illustrates SHIELD's effect on the POPE COCO subset with ambiguous inputs, showing that by leveraging noise-derived tokens to remove inaccurate representations, SHIELD significantly reduces hallucinations of dominant objects in pretraining data. Finally, Figure 8c highlights SHIELD's robustness against perturbations, where SHIELD-enhanced LLaVA-1.5 exhibits substantially less performance degradation under increasing attack steps.

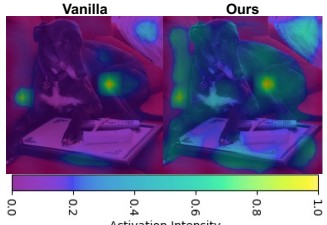
(a) Token Re-weighting

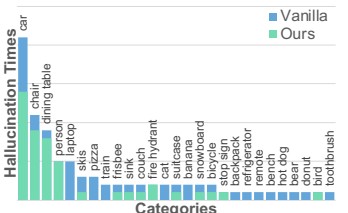
(b) Erroneous Representation Removal

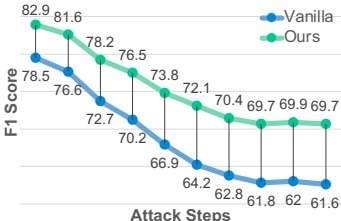
(c) Address Vulnerability

Figure 8: (a) Token Re-weighting highlights more object-relevant tokens. (b) Token Subtraction removes erroneous representations, reducing hallucinations associated with dominant objects in the pretraining data. Blue-only categories indicate zero hallucinations with our method. (c) Contrastive Decoding improves robustness to perturbations, mitigating vulnerability-induced hallucinations.

Table 8: Efficiency Comparison on CHAIR

| Method | $C_S\downarrow$ | T (s/sample)↓ | Mem↓ |
|---|---|---|---|
| Vanilla | 48.8 | 2.59 | 15.69GB |
| VCD | 46.8 | 4.89 | 16.52GB |
| OPERA | 44.6 | 24.01 | 34.88GB |
| Ours | **36.6** | 7.34 | 18.17GB |

Table 9: Module Efficiency on CHAIR

| Module | T (s/sample)↓ | Mem↓ |
|---|---|---|
| Vanilla | 2.59 | 15.69GB |
| w/ Mitigate Statistical Bias | 4.64 | 16.56GB |
| w/ Reduce Inherent Bias | 2.63 | 16.50GB |
| w/ Address Vulnerability | 7.30 | 18.17GB |
| Ours (All Modules) | 7.34 | 18.17GB |

**Inference Overhead Analysis.** To assess the computational cost of SHIELD, we evaluate its hallucination ratio, runtime, and peak GPU memory usage on the CHAIR benchmark with LLaVA-1.5 7B. As shown in Table 8, SHIELD achieves the lowest hallucination ratio ($C_S = 36.6$) while maintaining acceptable inference overhead. In particular, it runs substantially faster than OPERA and provides a more favorable trade-off between hallucination reduction and computational efficiency compared to VCD. To further analyze the sources of computational cost, we conduct a module-wise ablation study using the CHAIR benchmark and LLaVA-1.5 7B, as shown in Table 9. The majority of overhead arises from the Address Vulnerability and Mitigate Statistical Bias modules, both of which require caption generation. In particular, Address Vulnerability further involves adversarial tensor computation, increasing runtime to 7.30 seconds per sample and memory usage to 18.17 GB. The Mitigate Statistical Bias module, which only relies on caption generation, increases runtime to 4.64 seconds and memory to 16.56 GB. For reference, caption generation alone takes 2.05 seconds and 16.56 GB. In contrast, the Reduce Inherent Bias module introduces minimal overhead, adding only 0.04 seconds and 0.81 GB. Notably, the full SHIELD configuration incurs negligible additional cost beyond Address Vulnerability. These results indicate that most overhead is concentrated in caption generation and adversarial tensor computation, and that the overall trade-off can be flexibly adjusted by tuning caption length or the number of adversarial optimization steps.

## 5 CONCLUSION

This paper investigates object hallucinations in LVLMs, tracing their origin to visual encoders. Despite large-scale pretraining, these encoders suffer from three issues: statistical bias, which overemphasizes frequent patterns and distorts fine-grained perception; inherent bias, which induces erroneous representations related to dominant objects in pretraining data; and vulnerability, which makes encoders sensitive to minor perturbations and results in inaccurate features. To address these challenges, we propose SHIELD, a training-free framework that integrates token re-weighting, token subtraction, and contrastive decoding. Token re-weighting alleviates statistical bias by distributing attention to more ground-truth-relevant tokens. Token subtraction mitigates inherent bias by estimating and removing erroneous dominant-object representations using noise-derived tokens. Contrastive decoding counters vulnerability by exposing hallucinations via perturbed image and suppressing them through contrast with natural inputs. Extensive experiments demonstrate that SHIELD not only achieves significant improvements on hallucination benchmarks but also enhances general perception tasks, highlighting its effectiveness and broad applicability.

Future work could explore the design of more robust visual encoders to further reduce bias and vulnerability, as well as the optimization of adversarial attack mechanisms to lower computational costs for real-time applications.

ETHICS STATEMENT

In this paper, we propose SHIELD, a method that mitigates bias and vulnerability to reduce hallucinations in LVLMs, thereby enhancing their safety and reliability for the community. Outputs of SHIELD may occasionally contain inappropriate content inherited from the base model, and such outputs do not reflect the authors' views. We strictly adhere to the ICLR ethical research standards and applicable laws. To the best of our knowledge, this work complies with the General Ethical Principles.

REPRODUCIBILITY STATEMENT

We follow the ICLR reproducibility standards and ensure the reproducibility of our work. All datasets used for inference and evaluation are publicly available, and detailed experimental settings, including hyper-parameters and implementation steps, are documented in the paper and Appendix. Furthermore, we will release our code upon acceptance, enabling other researchers and practitioners to easily reproduce and extend our results.

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

Table 10: The Prompt used for GPT4o-aid evaluation

---

**GPT-4o Prompt**

You are required to score the performance of four AI assistants in describing a given image. You should pay extra attention to the hallucination, which refers to the part of descriptions that are inconsistent with the image content, such as claiming the existence of something not present in the image or describing incorrectly in terms of the counts, positions, or colors of objects in the image. Please rate the responses of the assistants on a scale of 1 to 10, where a higher score indicates better performance, according to the following criteria:
1: **Correctness:** whether the response is accurate with respect to the image content. Responses with fewer hallucinations should be given higher scores.
2: **Detailedness:** whether the response is rich in necessary details. Note that hallucinated descriptions should not count as necessary details.
Please output the scores for each criterion, containing only four values indicating the scores for Assistant 1, 2, 3 and 4, respectively. The four scores are separated by a space. Following the scores, please provide an explanation of your evaluation, avoiding any potential bias and ensuring that the order in which the responses were presented does not affect your judgment.
**[Assistant 1]**
{}
**[End of Assistant 1]**
**[Assistant 2]**
{}
**[End of Assistant 2]**
**[Assistant 3]**
{}
**[End of Assistant 3]**
**[Assistant 4]**
{}
**[End of Assistant 4]**
**Output format:**
Correctness: <Scores of the four answers>
Reason:
Detailedness: <Scores of the four answers>
Reason:

---

# A DETAILED EXPERIMENTAL SETUP

## A.1 GPT-4O ASSISTED EVALUATION

Following (Yin et al., 2023), we use GPT-4o to evaluate vanilla LVLM, VCD, OPERA, and our proposed SHIELD. Given a LVLM and an image, descriptions are generated using the prompt, "Please describe this image in detail". Using the evaluation prompt shown in Table 10, GPT-4o rates the four descriptions on a scale of 0 to 10 across two aspects: Correctness, which measures the consistency between the description and the image, assigning higher scores to highly consistent descriptions and lower scores to those with hallucinations, and Detailedness, which evaluates how comprehensively the description captures image details. The prompt instructs GPT-4o to disregard biases from sequential order and focus on inconsistencies, such as objects mentioned but absent in the image, including incorrect colors, positions, or relationships. Leveraging its analytical capabilities, GPT-4o performs a thorough and detailed evaluation.

Table 11: Comparison on POPE A-OKVQA and GQA Subset using LLaVA-1.5

| Dataset | Method | Random | | Popular | | Adversarial | | Average | |
|---------|--------|----------|------|----------|------|-------------|------|----------|------|
| | | Accuracy↑ | F1↑ | Accuracy↑ | F1↑ | Accuracy↑ | F1↑ | Accuracy↑ | F1↑ |
| AOKVQA | Vanilla | 83.4 | 82.5 | 79.9 | 79.5 | 74.0 | 75.1 | 79.1 | 79.0 |
| | VCD | 86.1 | 86.3 | 81.8 | 82.8 | 74.9 | 77.7 | 80.9 | 82.2 |
| | OPERA | 88.1 | 88.5 | 83.3 | 84.5 | 73.8 | 77.7 | 81.7 | 83.5 |
| | Ours | **90.3** | **89.9** | **85.3** | **85.5** | **77.2** | **79.1** | **84.2** | **84.8** |
| GQA | Vanilla | 83.7 | 82.9 | 78.1 | 78.3 | 75.0 | 76.0 | 78.9 | 79.0 |
| | VCD | 86.6 | 86.9 | 80.7 | 82.2 | 76.0 | 78.7 | 81.1 | 82.6 |
| | OPERA | 88.6 | 89.1 | 79.8 | 82.1 | 75.0 | 78.8 | 81.1 | 83.3 |
| | Ours | **90.5** | **90.3** | **84.2** | **84.8** | **79.5** | **81.0** | **84.7** | **85.3** |

# B MORE RESULTS

## B.1 POPE EVALUATION ON AOKVQA & GQA

To further validate the effectiveness of SHIELD, we conducted experiments on POPE using AOKVQA and GQA datasets under random, popular, and adversarial settings with the LLaVA-1.5

7B model. As shown in Tables 11, SHIELD significantly reduces hallucinations compared to previous methods. On average, SHIELD achieves an absolute improvement of 2.5 in Accuracy and 1.3 in F1 score on AOKVQA, and 3.6 in Accuracy and 2.0 in F1 score on GQA. Notably, SHIELD is particularly effective under the challenging Adversarial setting, underscoring biases and vulnerability in visual encoders as key contributors to object hallucination in these scenarios.

Table 12: Results on MME perception-related tasks using LLaVA-1.5

| Method | Existence | Count | Position | Color | Posters | Celebrity | Scene | Landmark | Artwork | OCR |
|--------|-----------|-------|----------|-------|---------|-----------|-------|----------|---------|-----|
| Vanilla | 175.67 | 124.67 | 114.00 | 151.00 | 127.82 | 113.59 | 148.30 | 129.95 | 102.20 | 92.00 |
| OPERA | 180.67 | 133.33 | 123.33 | 155.00 | 136.39 | **128.53** | **154.25** | **154.25** | **122.25** | 125.00 |
| VCD | 184.66 | 138.33 | 128.67 | 153.00 | 132.11 | 120.94 | 152.20 | 140.45 | 109.60 | 104.00 |
| Ours | **195.00** | **141.67** | **148.33** | **183.33** | **139.46** | 118.24 | 153.25 | 140.50 | 118.25 | **135.00** |

Table 13: Results on MME cognition-related tasks using LLaVA-1.5

| Method | Common Sense Reasoning | Numerical Calculation | Text Translation | Code Reasoning |
|--------|------------------------|-----------------------|------------------|----------------|
| Vanilla | 106.43 | **72.50** | **95.50** | 78.50 |
| OPERA | 114.29 | 40.00 | 87.50 | 62.50 |
| VCD | 111.29 | 68.50 | 89.50 | **84.00** |
| Ours | **122.86** | 57.50 | 82.50 | 75.00 |

## B.2 DETAILED RESULTS ON MME

Table 12 presents the results on perception-related tasks of the MME benchmark using the LLaVA-1.5 7B model. Compared to the vanilla LVLM, SHIELD achieves overall improvements, highlighting its ability to reduce hallucinations and enhance perception capabilities. This improvement likely stems from SHIELD's effectiveness in mitigating biases and vulnerabilities, thereby recalibrating the LVLM's visual feature extraction. When compared to previous methods, SHIELD achieves a higher total perception score (Table 6) but exhibits limited improvements on tasks requiring external knowledge beyond the input image (e.g., Celebrity, Scene, Landmark, Artwork). This limitation may result from contrastive decoding in SHIELD, which directs the LVLM to prioritize visual inputs over leveraging prior knowledge embedded in the LLM.

Furthermore, Table 13 showcases the results on cognition-related tasks within the MME benchmark using the LLaVA-1.5 7B model. The results show that applying SHIELD improves the LVLM's recognition on complex visual scenes (e.g., Common Sense Reasoning) but performs poorly on simpler visual scenes (e.g., Numerical Calculation, Text Translation, and Code Reasoning). This may be because simpler visual inputs are less likely to induce hallucinations, while contrastive decoding in SHIELD may limit the utilization of prior knowledge embedded in the LLM.

Table 14: GPT-4 Assisted Evaluation on MMHal-Bench

| Method | Overall ↑ | Hallucination ↓ | Attr. ↑ | Adv. ↑ | Comp. ↑ | Cnt. ↑ | Rel. ↑ | Env. ↑ | Hol. ↑ | Other ↑ |
|--------|-----------|-----------------|---------|--------|---------|--------|--------|--------|--------|---------|
| Vanilla | 2.21 | 0.70 | **2.75** | **2.00** | 2.50 | 2.50 | 1.67 | 2.17 | 1.83 | 2.25 |
| Ours | **2.65** | **0.61** | 2.42 | 1.83 | **2.58** | **2.67** | **2.83** | **3.92** | **2.08** | **2.83** |

## B.3 GPT-4 ASSISTED EVALUATION ON MMHAL-BENCH

Table 14 presents results on MMHal-Bench (Sun et al., 2024b), a benchmark designed to evaluate hallucinations in open-ended multimodal tasks. It spans eight hallucination-related categories, including attribute, adversarial object, comparison, counting, relation, environment, holistic description, and other visually grounded errors. To ensure up-to-date evaluation, we re-assess both the vanilla LLaVA-1.5 7B and our method using the latest version of GPT-4, as the original benchmark relied on the deprecated `gpt-4-0314`. Our method significantly reduces hallucination and improves performance across most categories, with the most notable gains observed in relation and environment understanding.

Table 15: Extended Evaluation on Recent VLMs Variants (POPE COCO Adversarial Split)

| Method | Qwen2-VL (7B) | | Qwen2.5-VL (7B) | | Qwen3-VL (8B) | |
|---|---|---|---|---|---|---|
| | Accuracy ($\uparrow$) | F1 Score ($\uparrow$) | Accuracy ($\uparrow$) | F1 Score ($\uparrow$) | Accuracy ($\uparrow$) | F1 Score ($\uparrow$) |
| Vanilla | 85.3 | 84.5 | 85.6 | 85.1 | 86.2 | 85.8 |
| Ours | **86.1** | **85.4** | **86.8** | **86.0** | **87.4** | **86.6** |

## B.4 EXTENDED EVALUATION ON RECENT VLMS

The VLMs used in our main experiments (LLaVA-1.5, InstructBLIP, and Qwen-VL) follow standard practice in recent multimodal hallucination studies, ensuring fair and reproducible comparison. To further evaluate the generalization of SHIELD, we extend our experiments to more recent Qwen-VL variants: Qwen2-VL (7B), Qwen2.5-VL (7B), and the newly released Qwen3-VL (8B). Notably, Qwen3-VL 8B was released in November 2025 and represents the latest generation of the Qwen-VL family. We report results on the POPE COCO adversarial split.

As shown in Table 15, SHIELD consistently improves both accuracy and F1 score across all models. These results demonstrate the effectiveness of our approach on more capable and recently released VLMs.

Table 16: Comparison with Training-Free and Training-Based Methods on CHAIR (512 tokens)

| Method | Training | $C_S(\downarrow)$ | $C_I(\downarrow)$ |
|---|---|---|---|
| Vanilla | ✗ | 48.8 | 14.2 |
| DoLa (Chuang et al., 2024) | ✗ | 47.7 | 13.8 |
| ICD (Wang et al., 2024) | ✗ | 47.4 | 13.9 |
| VCD (Leng et al., 2024) | ✗ | 46.8 | 13.2 |
| TAME (Tang et al., 2025) | ✗ | 45.2 | 14.0 |
| OPERA (Huang et al., 2024) | ✗ | 44.8 | 12.8 |
| SID (Huo et al., 2025) | ✗ | 44.2 | 12.2 |
| LLaVA-RLHF (Sun et al., 2024b) | ✓ | 43.6 | 10.5 |
| CCA-LLaVA (Xing et al., 2024) | ✓ | 43.0 | 11.5 |
| Less is More (Yue et al., 2024) | ✓ | 40.2 | 12.3 |
| MCA-LLaVA (Zhao et al., 2025b) | ✓ | 38.0 | 10.9 |
| SHIELD (Ours) | ✗ | **36.6** | **10.3** |

## B.5 EXTENDED COMPARISON WITH EXISTING HALLUCINATION MITIGATION METHODS

To provide a more comprehensive evaluation, we expand our comparison to include a wider set of hallucination mitigation methods on the CHAIR benchmark. These include both *training-free* methods (DoLa, ICD, VCD, TAME, OPERA, SID, and Ours) and *training-required* approaches (LLaVA-RLHF, CCA-LLaVA, Less is More, and MCA-LLaVA). All methods are implemented with LLaVA-1.5 7B to ensure consistency in backbone and evaluation setup.

As shown in Table 16, our method outperforms all training-free baselines and remains competitive with training-based methods. Notably, SHIELD achieves the lowest hallucination rates on both sentence and instance levels while requiring no additional VLM fine-tuning.

## C ADDITIONAL HYPER-PARAMETERS ABLATION

### C.1 EFFECT OF $\alpha$ IN CONTRASTIVE DECODING

Table 17 presents the results of an ablation study on $\alpha$, which controls the impact of contrastive decoding combined with adversarial attacks. The study shows a significant reduction in hallucinations on the CHAIR benchmark as $\alpha$ increases from 1.0 to 2.0, demonstrating the effectiveness of addressing vulnerabilities.

### C.2 EFFECT OF $\beta$ IN ADAPTIVE PLAUSIBLE CONSTRAINT

Table 18 presents the results of an ablation study on $\beta$, which controls the adaptive plausibility constraint. A larger $\beta$ indicates more aggressive truncation, retaining only high-probability tokens. As

Table 17: $\alpha$ Ablation

| $\alpha$ | $C_S\downarrow$ | $C_I\downarrow$ |
|---|---|---|
| 1.0 | 41.6 | 11.6 |
| 1.5 | 40.2 | 11.2 |
| 2.0 | **36.6** | **10.3** |
| 2.5 | 38.4 | 10.7 |

Table 18: $\beta$ Ablation

| $\beta$ | $C_S\downarrow$ | $C_I\downarrow$ |
|---|---|---|
| 0.20 | 36.8 | 11.2 |
| 0.25 | 38.0 | 11.0 |
| 0.30 | **36.2** | **10.3** |
| 0.35 | 36.6 | 10.3 |

Table 19: $K$ Ablation

| $K$ | $C_S\downarrow$ | $C_I\downarrow$ |
|---|---|---|
| 8 | 39.6 | 11.3 |
| 16 | 38.2 | 10.8 |
| 32 | **36.6** | **10.3** |
| 64 | 38.4 | 11.5 |

Table 20: $l$ Ablation

| $l$ | $C_S\downarrow$ | $C_I\downarrow$ |
|---|---|---|
| 0.01 | 37.8 | 11.2 |
| 0.02 | **36.6** | **10.3** |
| 0.03 | 38.0 | 11.2 |
| 0.04 | 43.2 | 11.6 |

$\beta$ increases, changes in hallucination reduction exhibit minor fluctuations, suggesting that while the adaptive plausibility constraint, as an integral part of contrastive decoding, prevents the generation of implausible content, it plays a limited role in alleviating object hallucinations.

### C.3 EFFECT OF $K$ IN REDUCING INHERENT BIAS

Table 19 presents the results of an ablation study on $K$, which specifies the number of noise inputs used to estimate inherent bias for subsequent removal from visual tokens. Increasing $K$ improves the accuracy of inherent bias estimation, resulting in more effective hallucination mitigation. However, when $K$ becomes excessively large, the estimated bias converges toward zero, limiting its impact on mitigating hallucinations.

### C.4 EFFECT OF $l$ IN ADDRESSING VULNERABILITY

Table 20 presents the results of an ablation study on the learning rate within Vulnerability Addressing, which controls the granularity of adversarial attack tensor computation. When the learning rate is too large, the generated attack tensor fails to adapt effectively to image details, reducing its effectiveness. Consequently, the subsequent contrastive decoding process cannot adequately minimize hallucinations caused by vulnerabilities in the visual encoder.

### C.5 EFFECT OF ATTACK STEP IN ADDRESSING VULNERABILITY

Table 21 presents an ablation study on the number of adversarial optimization steps in addressing vulnerability module. This parameter controls how thoroughly the attack tensor explores encoder weaknesses. The results show that performance saturates at 30 steps, with no further gains at 45 steps.

Table 21: Attack Steps Ablation

| Steps | $C_S\downarrow$ | $C_I\downarrow$ |
|---|---|---|
| 15 | 40.8 | 11.6 |
| 30 | **36.6** | **10.3** |
| 45 | 36.6 | 10.4 |

## D ADDITIONAL ANALYSIS

Table 22: Class-wise hallucination rate (%) under different noise settings on POPE COCO

| Input | person | skis | snowboard | bird | backpack | skateboard | bowl | scissors | knife | keyboard | carrot | refrigerator | remote | sandwich | sink |
|---|---|---|---|---|---|---|---|---|---|---|---|---|---|---|---|
| Noise Img | 100 | 80 | 30 | 30 | 20 | 20 | 20 | 10 | 10 | 10 | 10 | 10 | 10 | 10 | 0 |
| Noise Tok | 0 | 10 | 10 | 0 | 10 | 0 | 0 | 10 | 0 | 10 | 0 | 0 | 10 | 0 | 20 |

### D.1 ADDITIONAL INHERENT BIAS ANALYSIS

To further examine inherent bias in the visual encoder, we conduct a controlled experiment using the POPE COCO subset. For each object category, we design 10 identical POPE-style questions (e.g., "Is there a {object} in the image?") and evaluate hallucination rates under two distinct noise settings. In the first setting, the visual encoder receives noise as image input (Noise Image), while in the second, the original image is retained but its visual tokens are replaced with noise embeddings (Noise Visual Tokens).

We measure the proportion of "Yes" responses, which indicate hallucination, for each object. As shown in Table 22, hallucinations under the Noise Image setting are heavily skewed, with certain categories (e.g., person) being disproportionately predicted. In contrast, hallucination rates under Noise Visual Tokens remain uniformly across categories. These findings confirm that the visual

encoder exhibits class-specific inherent biases, which can be effectively revealed through noise-based evaluation.

To mitigate the impact of randomness from individual noise inputs, SHIELD averages responses over $K$ noise inputs. This aggregation helps reduce variance and provides a more stable estimate of the encoder's bias tendencies.

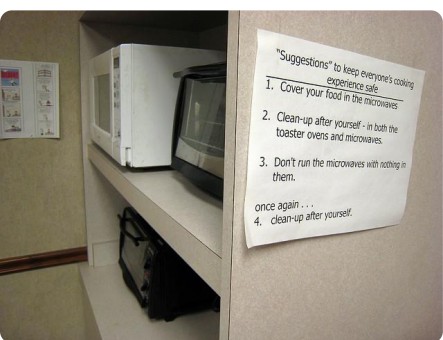

**Raw Caption:**
In the image, there is a microwave oven sitting on top of a counter. The microwave is placed next to a toaster oven, and both appliances appear to be clean and well-maintained. Above the microwave and toaster oven, there is a sign with suggestions to keep everyone

**Hallucinated Caption:**
The image features a kitchen with a microwave sitting on top of a counter. The microwave is placed on a shelf, and there is a sign above it, possibly providing instructions or suggestions. The kitchen also has a refrigerator, which is located on the left side of the scene.

Figure 9: Example naive captions with and without plausible hallucination. The hallucinated version includes a refrigerator not present in the image.

Table 23: Impact of Plausible Hallucinations in Naive Caption

| Model | Accuracy ($\uparrow$) | F1-Score ($\uparrow$) |
|---|---|---|
| Vanilla | 78.90 | 77.50 |
| Ours w/ Raw Caption | **80.13** | **79.14** |
| Ours w/ Hallucinated Caption | 79.83 | 78.81 |

## D.2 Plausible Hallucination in Naive Caption Analysis

To assess the effect of plausible hallucinations in naive captions, we conduct a controlled experiment on the POPE COCO adversarial subset. We simulate plausible hallucinations by introducing mild noise into the visual input during naive caption generation. This can produce hallucinated objects that are contextually reasonable but not actually present in the image.

Figure 9 shows one representative example where the hallucinated caption mentions a "refrigerator", which is plausible in the context but does not appear in the image. Despite this, grounded elements such as "microwave" and "sign" remain in the caption, which continue to support effective token re-weighting. The impact of such plausible hallucinations is limited because they typically have lower visual grounding and do not dominate the token distribution.

To quantify this, we compare SHIELD's performance when using raw versus hallucinated captions. As shown in Table 23, the drop in performance is minor, indicating robustness of SHIELD to plausible caption noise.

Table 24: Effect of Attack Strategy on CHAIR

| Configuration | $C_S$ ($\downarrow$) | $C_I$ ($\downarrow$) |
|---|---|---|
| Vanilla | 48.8 | 14.2 |
| Ours w/ FGSM-based Vulnerability Module | 39.2 | 11.5 |
| Ours w/ PGD-based Vulnerability Module | 37.2 | 11.3 |
| Ours (Learnable Attack) | **36.6** | **10.3** |

## D.3 Attack Strategy Analysis

To evaluate the design of our vulnerability addressing module, we compare our learnable perturbation with standard adversarial attacks that use fixed step sizes, such as FGSM (Goodfellow et al., 2015) and PGD (Madry et al., 2018). Instead of relying on fixed-magnitude updates, our method

learns a fine-grained perturbation through adversarial optimization between the visual encoder and its paired CLIP text encoder (see Section 3.2.4). Conditioned on both the image and naive caption, the perturbation is specifically optimized to expose vulnerabilities in the encoder, offering more flexible and precise control than fixed-step-size attacks.

To validate its effectiveness, we evaluate multiple configurations on the CHAIR benchmark, including variants where the vulnerability addressing module is implemented with FGSM or PGD instead. As shown in Table 24, while all configurations incorporating the vulnerability module help reduce hallucinations, our learnable attack achieves the lowest scores on both $C_S$ and $C_I$.

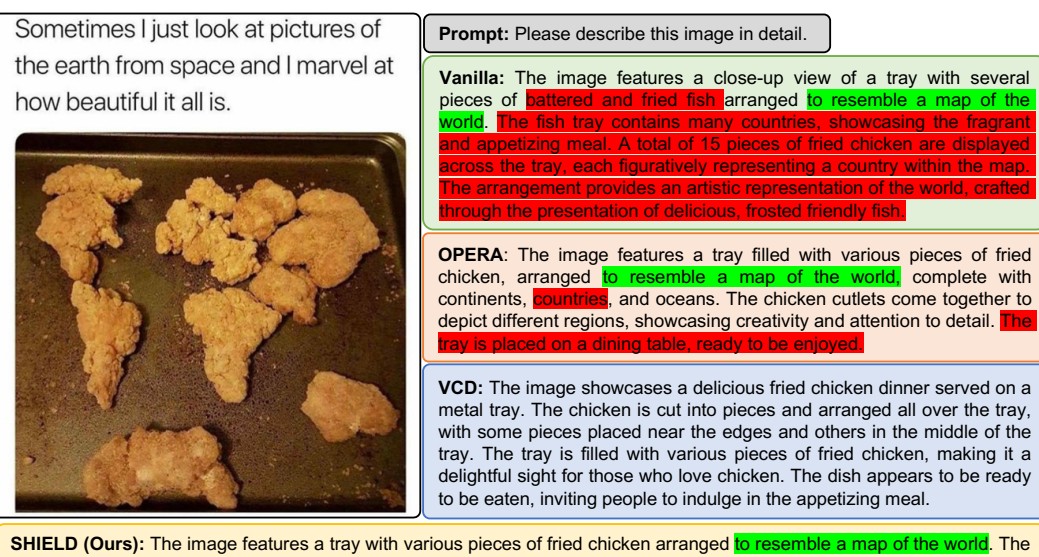

(a) Case Study on LLaVA-Bench No.14

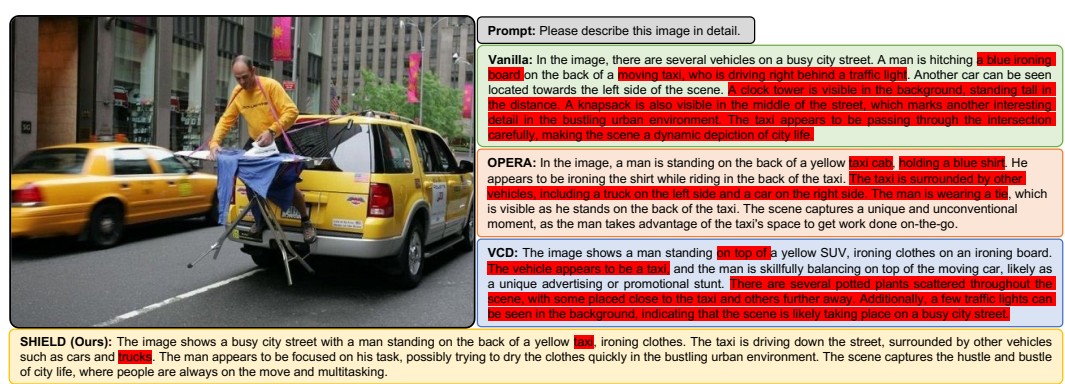

(b) Case Study on LLaVA-Bench No.10

Figure 10: Examples from LLaVA-Bench demonstrate the effectiveness of our method in correcting hallucinations. Hallucinated content is highlighted in red, and key information is highlighted in green.

## E  CASE STUDY

Figures 10 present two case studies demonstrating how vanilla LVLMs and previous methods, given identical prompts and images, can produce object hallucinations due to biases and vulnerabilities in the visual encoder. For instance, in Figure 10a, shadows and blurriness along the tray's edges expose

visual encoder vulnerabilities, leading the vanilla LVLM to misidentify fried chicken as fried fish. Similarly, in Figure 10b, statistical bias causes the vanilla LVLM to overemphasize tokens associated with frequent visual concepts in CLIP's pre-training data, thereby distorting detail perception and incorrectly identifying the ironing board as being the same blue color as the shirt. In contrast, SHIELD effectively mitigates hallucinations while preserving the coherence and informativeness of the generated text.

## F    THE USE OF LARGE LANGUAGE MODELS (LLMS)

We used GPT for two purposes: (i) polishing grammar and improving readability, and (ii) assisting in the evaluation of LVLM outputs. All research ideas and analyses were conducted by the authors, who take full responsibility for the content.

