# OpenReview forum: "SHIELD: Suppressing Hallucinations In LVLM Encoders via Bias and Vulnerability Defense"
_ICLR.cc/2026/Conference — ICLR 2026 Poster_

### Official Review · Reviewer_AwNy · 2025-10-27

**Soundness:** 4
**Presentation:** 3
**Contribution:** 4
**Rating:** 8
**Confidence:** 4

**Summary:**

This paper identifies three encoder-level sources of object hallucination—statistical bias, inherent bias, and vulnerability—and proposes SHIELD, a training-free pipeline that integrates token re-weighting, token subtraction (noise-derived removal of erroneous features), and adversarially informed contrastive decoding to mitigate these issues. Extensive experiments on CHAIR / POPE / MME evaluations demonstrate consistent gains across several LVLM families.

**Strengths:**

1. Clear identification of encoder-level failure modes. The decomposition into statistical bias, inherent bias, and vulnerability is conceptually clean and well motivated by analyses (token peak-to-average ratios, dominant-object errors under noise, perturbation sensitivity).

2. Strong token-level analysis. The paper’s quantitative and visualization evidence showing how individual visual tokens (peak norms / overemphasis) correlate with hallucination risk is compelling and valuable: it grounds the solution at the token representation level rather than treating hallucination as a black-box LLM issue. This token perspective is an important contribution for future encoder diagnostics.

3. Modular, training-free design. SHIELD’s three modules are practical (can be combined or ablated) and the paper provides ablations showing complementary gains. Precomputable components (e.g., noise-derived estimates) make the approach more practical.

**Weaknesses:**

1.	Generative / open-ended hallucination benchmarks.
The current evaluation focuses mainly on binary or existence-based benchmarks (POPE, MME). To better demonstrate robustness under real generative scenarios, the paper should include results on open-ended hallucination benchmarks such as AMBER [1], MMHal-Bench [2], which assess caption-level fidelity and factual consistency rather than yes/no predictions.

2.	Comparison with token pruning or sparse-token methods.
It would be useful to compare SHIELD with token pruning or sparse-token selection approaches that also modify the visual token set before decoding.

3.	Ablation on precomputation and inference overhead.
Since the paper mentions precomputing noise-derived erroneous representations, reporting runtime and memory trade-offs between precomputation and online computation would make the efficiency claims more convincing.

4.	Broader hallucination metrics and human evaluation.
For open-ended captioning, automatic metrics can be unreliable. Including limited human or GPT-4–based evaluation [2] focusing on whether hallucinated objects are introduced or removed would enhance credibility and interpretability.

[1] Amber: An llm-free multi-dimensional benchmark for mllms hallucination evaluation, 2023.

[2] Aligning large multimodal models with factually augmented rlhf, ACL 2024.

**Questions:**

See weakness

**Details Of Ethics Concerns:**

No concern

---

> ### Author Response · Authors · 2025-11-20
> **Response to Reviewer AwNy (Part 1)**
>
> ## Response to Reviewer AwNy
>
> We sincerely thank Reviewer AwNy for the thoughtful comments and constructive evaluation. Below, we address each of the concerns in detail.
>
> ### Q1 & Q4. Generative / Open-ended Hallucination Benchmarks and Metrics
>
> > *“The current evaluation focuses mainly on binary or existence-based benchmarks...”*
> > *“For open-ended captioning, automatic metrics can be unreliable...”*
>
> We appreciate the reviewer's suggestion. To address this concern, we extended our evaluation to two hallucination benchmarks: AMBER [1] and MMHal-Bench [2].
>
> **AMBER** provides both **generative** and **discriminative** hallucination metrics, including CHAIR, coverage, hallucination rate, cognitive effort, and standard classification scores:
>
> **Table – AMBER Benchmark Evaluation (LLaVA-1.5 7B)**
> ↓ = Lower is better ↑ = Higher is better
> *Generative*: CHAIR ↓, Cover ↑, Hal ↓, Cog ↓
> *Discriminative*: Acc ↑, Precision ↑, Recall ↑, F1 ↑
>
> | Method  | CHAIR ↓ | Cover ↑ | Hal ↓ | Cog ↓ | Acc ↑ | Precision ↑ | Recall ↑ | F1 ↑  | Amber Score ↑ |
> |---------|---------|---------|--------|--------|--------|--------------|----------|--------|----------------|
> | Vanilla | 9.2     | 41.3    | 29.2   | 3.7    | 65.7   | 83.2         | 64.7     | 73.2   | 82.0           |
> | VCD[3]     | 8.1     | 44.2    | 28.6   | 3.1    | 68.3   | 85.8         | 65.2     | 74.0   | 82.9           |
> | OPERA[4]   | 8.3     | 43.1    | 31.2   | 2.9    | 76.0   | 79.2         | 83.8     | 81.4   | 86.5           |
> | Ours    | **6.4** | **46.1**| **25.1**| **1.8**| **78.3**| **89.1**   | 76.6     | **82.4** | **88.0**       |
>
> Our method achieves the highest Amber Score (88.0) and consistently outperforms all LLaVA-1.5 7B based baselines across most metrics.
>
> For **MMHal-Bench**, we note that the original benchmark used the now-deprecated `gpt-4-0314` model. To ensure up-to-date results, we re-evaluate both the vanilla LLaVA-1.5 7B model and our method using the latest GPT-4 version.
>
> **Table – MMHal-Bench Evaluation (LLaVA-1.5 7B)**
> ↑ = Higher is better ↓ = Lower is better (Hallucination Rate)
> | Method  | Overall ↑ | Hallucination ↓ | Attribute ↑ | Adversarial ↑ | Comparison ↑ | Counting ↑ | Relation ↑ | Environment ↑ | Holistic ↑ | Other ↑ |
> |---------|-----------|------------------|--------------|----------------|---------------|--------------|--------------|----------------|--------------|--------|
> | Vanilla | 2.21      | 0.70             | **2.75**         | **2.00**       | 2.50          | 2.50         | 1.67         | 2.17           | 1.83         | 2.25   |
> | Ours    | **2.65**  | **0.61**         | 2.42         | 1.83           | **2.58**      | **2.67**     | **2.83**     | **3.92**       | **2.08**     | **2.83** |
>
> Our method significantly reduces hallucination rate and improves performance across most categories, especially in **relation** and **environment** understanding.
>
>
>
>
>
> ### Q2. Comparison with Token Pruning or Sparse-token Methods
>
> > *“Comparison with token pruning or sparse-token methods. It would be useful to compare SHIELD with token pruning or sparse-token selection approaches that also modify the visual token set before decoding.”*
>
> We thank the reviewer for this insightful suggestion. Based on LLaVA-1.5 7B, we compare SHIELD with token pruning methods on the MME benchmark, which evaluates a broad range of multimodal understanding capabilities. The table below presents both the MME scores and the number of visual tokens for pruning-based methods (ATP-LLaVA, PruMerge) and hallucination mitigation methods (OPERA, VCD, ours).
>
> **Table - MME Score and Visual Token Count**
>
> | Method             | MME Score ↑ | Number of Visual Tokens ↓ |
> |--------------------|-------------|-----------------------------|
> | Vanilla            | 1632        | 576                         |
> | ATP-LLaVA[5]       | 1474        | 144                         |
> | LLaVA-PruMerge[6]  | 1462        | 144                         |
> | OPERA[4]           | 1717        | 576                         |
> | VCD[3]             | 1717        | 576                         |
> | Ours               | **1810**    | 576                         |
>
> Pruning methods aim to reduce the number of visual tokens while maintaining performance. In contrast, hallucination mitigation methods do not reduce visual tokens, but focus on improving the reliability of the answers.
>
> Importantly, all hallucination mitigation methods outperform the vanilla model on MME. Our method achieves the highest overall score, suggesting that reducing hallucination not only improves faithfulness but also enhances general VLM performance.

---

> > ### Author Response · Authors · 2025-11-20
> > **Response to Reviewer AwNy (Part 2)**
> >
> > ### Q3. Inference Overhead
> >
> > > *“Ablation on precomputation and inference overhead ...”*
> >
> > Thank you for highlighting this important point. We agree that inference efficiency is critical for deployment. Below we report SHIELD's latency and memory usage compared to baselines on LLaVA-1.5 7B:
> >
> > **Table – Efficiency and Performance on CHAIR**
> > | Method  | Chair-S ↓ | Speed (s/sample) ↓ | Max GPU Memory ↓ |
> > |---------|------------|--------------------|------------------|
> > | Vanilla | 48.8       | 2.59               | 15.69GB          |
> > | VCD[3]  | 46.8       | 4.89               | 16.52GB          |
> > | OPERA[4]| 44.6       | 24.01              | 34.88GB          |
> > | Ours    | **36.6**   | 7.34               | 18.17GB          |
> >
> > SHIELD provides the best performance with acceptable overhead, which is significantly faster than OPERA, and more efficient than VCD relative to its larger gain.
> >
> > We also conduct a module-wise ablation:
> >
> > **Table – Module-wise Ablation**
> > | Configuration                                | Chair-S ↓ | Speed (s/sample) ↓ | Max GPU Memory ↓ |
> > |---------------------------------------------|-----------|---------|----------|
> > | Vanilla                                      | 48.8      | 2.59    | 15.69GB  |
> > | Vanilla + Mitigate Statistical Bias *(needs caption)* | 46.8      | 4.64    | 16.56GB  |
> > | Vanilla + Reduce Inherent Bias                       | 48.2      | 2.63    | 16.50GB  |
> > | Vanilla + Address Vulnerability *(needs caption & adv. tensor)* | 46.4 | 7.30 | 18.17GB  |
> > | Ours (All Modules)                           | **36.6**  | 7.34    | 18.17GB  |
> >
> > Most cost comes from generating the naive caption and adversarial tensor. For reference, caption generation alone takes **2.05 s/sample** and **16.56 GB**. Additionally, the tradeoff can be tuned by adjusting caption length or adversarial optimization steps.
> >
> >
> > References
> > [1] Wang J, Wang Y, Xu G, et al. Amber: An llm-free multi-dimensional benchmark for mllms hallucination evaluation. arXiv:2311.07397.
> > [2] Sun Z, Shen S, Cao S, et al. Aligning large multimodal models with factually augmented rlhf. ACL 2024.
> > [3] Leng S, Zhang H, Chen G, et al. Mitigating object hallucinations in large vision-language models through visual contrastive decoding. CVPR 2024.
> > [4] Huang Q, Dong X, Zhang P, et al. Opera: Alleviating hallucination in multi-modal large language models via over-trust penalty and retrospection-allocation. CVPR 2024.
> > [5] Ye X, Gan Y, Ge Y, et al. Atp-llava: Adaptive token pruning for large vision language models. CVPR 2025.
> > [6] Shang Y, Cai M, Xu B, et al. Llava-prumerge: Adaptive token reduction for efficient large multimodal models. CVPR 2025.

---

> > > ### Comment · Reviewer_AwNy · 2025-11-21
> > >
> > > Thanks for the detailed response. The added experiments on AMBER and MMHal-Bench are convincing and they effectively address my main concern regarding open-ended generation. The efficiency analysis and the comparison with token pruning are also clear to me now.
> > >
> > > Overall, this is a high-quality paper with strong novelty, particularly in its token-level decomposition of hallucination sources. I have no further questions and will maintain my score.

---

> > > > ### Author Response · Authors · 2025-11-21
> > > >
> > > > We thank the reviewer for the positive feedback and are glad that the additional experiments and analyses addressed your concerns. We appreciate your encouraging evaluation and thank you for maintaining your score.

---

### Official Review · Reviewer_85aF · 2025-10-28

**Soundness:** 3
**Presentation:** 3
**Contribution:** 3
**Rating:** 6
**Confidence:** 4

**Summary:**

This paper identifies the visual encoder as a source of object hallucinations in Large Vision-Language Models (LVLMs), suggesting this stems from three issues: statistical bias, inherent bias, and vulnerability. The authors propose SHIELD, a training-free, post-hoc framework to mitigate these issues. SHIELD consists of three modules: (1) Token Re-weighting, which uses a naively generated caption to up-weight relevant visual tokens; (2) Token Subtraction, which estimates and removes a "bias" feature derived from noise inputs; and (3) Contrastive Decoding, which penalizes logits that are sensitive to a small adversarial perturbation on the input image. Experiments on several LVLMs (LLaVA, InstructBLIP, Qwen-VL) and hallucination benchmarks (CHAIR, POPE) show that SHIELD can reduce hallucination rates compared to other training-free methods.

**Strengths:**

- The paper's focus on diagnosing hallucination sources within the visual encoder is a valuable contribution. The categorization into statistical bias, inherent bias, and vulnerability provides a structured way to think about the problem, moving beyond the more common focus on the LLM component.

- The design of SHIELD is logical, with each of its three components clearly mapped to one of the three identified problems. This modularity makes the approach easy to understand.

- As a post-hoc, training-free method, SHIELD offers a practical advantage, as it can be applied to any pre-trained LVLM without requiring expensive re-training or fine-tuning.

- The experiments are thorough, covering multiple model families and benchmarks. The results demonstrate that SHIELD consistently improves over the vanilla models and outperforms other training-free baselines like VCD and OPERA in reducing object hallucinations, especially on benchmarks like POPE.

**Weaknesses:**

1. The primary weakness is the method's high computational cost at inference time. SHIELD requires multiple additional forward passes: one to generate the "naive caption," $K$ (e.g., 32) passes for the noise inputs, and another pass for the adversarially attacked image. This combined overhead likely makes the method prohibitively slow for any practical or real-time application, which severely limits its impact.

2. The method introduces several new and sensitive hyperparameters ($\alpha$ for subtraction, $\beta$ for contrastive decoding, $K$ for noise samples, and the attack parameters $l$ and step size). The paper provides very little ablation or sensitivity analysis for these. It's unclear how these were tuned or how performance would change on a different dataset or model, making the results potentially difficult to reproduce or generalize.

3. The idea of "inherent bias" being estimated by feeding pure noise to the encoder is a strong assumption. It is not convincingly demonstrated that the resulting feature vector is a meaningful representation of "bias" rather than just an artifact of the network's response to out-of-distribution data. Furthermore, using a large $K=32$ to get a stable estimate adds significantly to the computational burden.

4. The "Token Re-weighting" module fully depends on an initial caption from the vanilla LVLM, which is itself prone to hallucination. The paper's claim that this doesn't matter (because hallucinated words won't match visual tokens) seems overly optimistic and is not well-supported. A plausible hallucination could easily bias the re-weighting in the wrong direction, but this potential failure mode is not investigated.

**Questions:**

1. Can the authors provide a quantitative analysis of the inference-time latency (e.g., in seconds per image) and memory overhead of SHIELD compared to the vanilla model and the baselines (VCD, OPERA)? How much does each of the three modules (re-weighting, subtraction, contrastive decoding) contribute to this overhead?

2. The paper lacks a sensitivity analysis for the key hyperparameters $\alpha$, $\beta$, and $K$. How much does performance change when varying these? How were the final values chosen?

3. Regarding the "Token Re-weighting" module: What happens if the "naive caption" contains a plausible hallucination (e.g., seeing a "boat" in a wide shot of a lake)? Can you provide an analysis of this failure case?

4. Why was the specific PGD-style attack chosen for the "Address Vulnerability" module? How does the method perform if a different type of attack is used (e.g., FGSM) or if no attack is used at all (i.e., only using modules 1 and 2)?

5. For the CHAIR evaluation, the paper follows prior work using 500 randomly selected images. How is the balance between coverage (ensuring all present objects are in the ground-truth) and faithfulness (ensuring no non-existent objects are in the ground-truth) managed for this label set? This seems critical for a metric that punishes “hallucinated objects”.

6. For the GPT-4o assisted evaluation, why was a 0-10 human-like scoring system for “correctness” and “detailedness” chosen? Given recent work on automated evaluation (e.g., Valor-eval: Holistic coverage and faithfulness evaluation of large vision-language models. https://arxiv.org/pdf/2404.13874), what are the advantages of this 0-10 scale over more objective, feature-extraction-based or model-based evaluation methods?

---

> ### Author Response · Authors · 2025-11-20
> **Response to Reviewer 85aF (Part 1)**
>
> ## Response to Reviewer 85aF
>
> We sincerely thank Reviewer 85aF for the detailed and constructive feedback. We address each of the points raised below.
>
> ### W1 & Q1. Computational Cost
>
> > *“The primary weakness is the method's high computational cost at inference time ...”*
> > *“Can the authors provide a quantitative analysis of the inference-time latency and memory overhead ...”*
>
> We appreciate the reviewer's concern regarding inference efficiency, which is indeed critical for practical deployment. Below, we provide a detailed comparison of SHIELD's runtime and memory usage on LLaVA-1.5 7B, using the CHAIR benchmark:
>
> **Table – Efficiency and Performance on CHAIR**
> | Method  | Chair-S ↓ | Speed (s/sample) ↓ | Max GPU Memory ↓ |
> |---------|------------|--------------------|------------------|
> | Vanilla | 48.8       | 2.59               | 15.69GB          |
> | VCD     | 46.8       | 4.89               | 16.52GB          |
> | OPERA   | 44.6       | 24.01              | 34.88GB          |
> | Ours    | **36.6**   | 7.34               | 18.17GB          |
>
> SHIELD provides the best performance with acceptable overhead, which is significantly faster than OPERA, and more efficient than VCD relative to its larger gain.
>
> To better understand the cost breakdown, we include a module-wise ablation:
>
> **Table – Module-wise Ablation**
> | Configuration                                | Chair-S ↓ | Speed (s/sample) ↓ | Max GPU Memory ↓ |
> |---------------------------------------------|-----------|---------|----------|
> | Vanilla                                      | 48.8      | 2.59    | 15.69GB  |
> | Vanilla + Mitigate Statistical Bias *(needs caption)* | 46.8      | 4.64    | 16.56GB  |
> | Vanilla + Reduce Inherent Bias                       | 48.2      | 2.63    | 16.50GB  |
> | Vanilla + Address Vulnerability *(needs caption & adv. tensor)* | 46.4 | 7.30 | 18.17GB  |
> | Ours (All Modules)                            | **36.6**  | 7.34    | 18.17GB  |
>
> Most cost comes from generating the naive caption and adversarial tensor. For reference, caption generation alone takes **2.05 s/sample** and **16.56 GB**. Additionally, the tradeoff can be tuned by adjusting caption length or adversarial optimization steps.
>
>
> Furthermore, we clarify that the $K$ noise inputs used in Reduce Inherent Bias are processed in a single batch through the visual encoder. As this step is input-agnostic (i.e., it depends only on the encoder architecture), the result can be precomputed once per model and reused, as described in Section 3.2.3 (Line 270).
>
>
>
> ### W2 & Q2. Hyperparameters Ablation
>
> > _“The method introduces several new and sensitive hyperparameters ...”_
> > _“The paper lacks a sensitivity analysis for the key hyperparameters $\alpha$, $\beta$, $K$ and $l$ ...”_
>
> Thank you for the suggestion. Ablation studies for the key hyperparameters $\alpha$, $\beta$, $K$, and $l$ are originally included in Appendix Section C “ADDITIONAL HYPER-PARAMETERS ABLATION” (Tables 12–15).
>
> In addition, we now include an ablation of the adversarial optimization step size on the CHAIR dataset (i.e., number of attack iterations) below:
>
> **Table – Ablation on Attack Steps**
> | Attack Steps | Chair-S ↓ | Chair-I ↓ |
> |--------------|-----------|-----------|
> | 15           | 40.8      | 11.6      |
> | 30           | **36.6**  | **10.3**  |
> | 45           | 36.6      | 10.4      |
>
> These results show that performance saturates at 30 steps, and further increasing the steps bring no additional benefit.

---

> ### Author Response · Authors · 2025-11-20
> **Response to Reviewer 85aF (Part 2)**
>
> ### W3. Inherent Bias Assumption
> > *“The idea of 'inherent bias' being estimated by feeding pure noise to the encoder is a strong assumption...”*
> > *“Furthermore, using a large K = 32 to get a stable estimate adds significantly to the computational burden...”*
>
> Thank you for raising this point. We appreciate the opportunity to clarify this component. To empirically verify our assumption, we conduct a controlled experiment on the POPE COCO subset, constructing 10 identical POPE-style questions per object (e.g., *“Is there a {object} in the image?”*) and comparing hallucination rates across two noise settings:
> 1. **Noise Image**: Feed pure noise to the visual encoder.
> 2. **Noise Visual Tokens**: Replace the visual tokens with noise embeddings.
>
> We then measure the percentage of “Yes” responses (i.e., hallucinations) for each object class. As shown in the table below, Noise Image leads to skewed hallucination distributions, with certain categories (e.g., person, skis, snowboard) being disproportionately hallucinated. In contrast, hallucinations from Noise Visual Tokens are uniformly low and flat. This supports our assumption that the encoder carries **object-specific inherent biases** that can be reliably revealed via noise inputs.
>
> **Table – Class-wise hallucination rate (%) under different noise settings**
> | Input              | person | skis | snowboard | bird | backpack | skateboard | bowl | scissors | knife | keyboard | carrot | refrigerator | remote | sandwich | sink |
> |---|---|---|---|---|---|---|---|---|---|---|---|---|---|---|---|
> | Noise Image        | 100 | 80 | 30 | 30 | 20 | 20 | 20 | 10 | 10 | 10 | 10 | 10 | 10 | 10 | 0  |
> | Noise Visual Tokens| 0   | 10 | 10 | 0  | 10 | 0  | 0  | 10 | 0  | 10 | 0  | 0  | 10 | 0  | 20 |
>
> To minimize the impact of outliers from individual noise samples, SHIELD average the outputs over all $K$ inputs. This aggregation smooths out noise-specific fluctuations and more clearly estimates the encoder's systematic bias tendencies.
>
> Importantly, as mentioned in response to W1 & Q1, all $K$ noise inputs are passed together in a single forward pass, and the results can be cached for reuse, further mitigating computational cost.
>
> ### W4 & Q3. Plausible Hallucination in Naive Caption
> > *“The Token Re-weighting module fully depends on an initial caption ... ”*
> > *“What happens if the naive caption contains a plausible hallucination?”*
>
> Thank you for the insightful question. Our claim in Line 256 that hallucinated objects do not affect token re-weighting mainly applies to implausible hallucinations.
>
> For plausible hallucinations, we note that the Token Re-weighting module mitigates statistical bias, which refers to the overemphasis on certain visual tokens, by guiding the model to attend to more tokens based on the naive caption. Therefore, when the caption includes a plausible hallucination (such as “boat”), the module tends to highlight contextually relevant tokens that are actually present in the image (such as “lake”), which is generally not harmful.
>
> Only when the highlighted tokens of a plausible hallucination overlaps with the already statistically biased tokens might the module's guidance be weakened. However, this effect is limited because naive captions typically include multiple grounded objects that still guide the re-weighting process effectively. Additionally, plausible hallucinated objects usually exhibit lower similarity to visual tokens compared to grounded objects in the caption, which further reduces their negative impact.
>
> To validate this, we simulate `plausible hallucinations` by adding mild noise to the visual input when generating the naive caption. Below is a concrete example:
>
> - **Raw caption**: “In the image, there is a microwave oven sitting on top of a counter. The microwave is placed next to a toaster oven, and both appliances appear to be clean and well-maintained. Above the microwave and toaster oven, there is a sign with suggestions.”
>
> - **Plausible hallucinated caption**: “The image features a `kitchen` with a microwave sitting on top of a counter. The microwave is placed on a shelf, and there is a sign above it, possibly providing instructions or suggestions. `The kitchen also has a refrigerator, which is located on the left side of the scene.`”
>
> The corresponding image paired with these captions will be included in the updated PDF to better illustrate the hallucination example.
>
> We then evaluate SHIELD on the POPE COCO adversarial split using both the raw and hallucinated captions:
>
> **Table – Impact of Plausible Hallucinations in Naive Caption**
> | Module | Acc ↑ | F1-Score ↑ |
> |---|---|---|
> | Vanilla | 78.90 | 77.50 |
> | SHIELD w/ Raw Caption | 80.13 | 79.14 |
> | SHIELD w/ Hallucinated Caption | 79.83 | 78.81 |
>
> Compared to using the Raw caption, the performance of Hallucinated Caption drops only slightly (–0.30% accuracy, –0.33% F1), which we consider an acceptable tradeoff given the method's robustness.

---

> > ### Author Response · Authors · 2025-11-20
> > **Response to Reviewer 85aF (Part 3)**
> >
> > ### Q4. Attack Method
> >
> > > *“Why was the specific PGD-style attack chosen for the "Address Vulnerability" module? How does the method perform if a different type of attack is used (e.g., FGSM) or if no attack is used at all (i.e., only using modules 1 and 2)?”*
> >
> > We thank the reviewer for this thoughtful question. We clarify that instead of using fixed step size attacks like PGD or FGSM, we adopt a **learnable perturbation** optimized via adversarial learning between the visual encoder and its paired CLIP text encoder as described in Section 3.2.4. This perturbation is derived from the original image and naive caption, and specifically learn to expose hallucinations rooted in the **vulnerability** of the visual encoder. Compared to PGD or FGSM, the learnable attack is more flexible and fine-grained, avoiding constraints from fixed step sizes.
> >
> > We agree that further comparison helps improve completeness. Below we provide results on the CHAIR dataset using different attack strategies, along with module ablation when only the bias-related components are used.
> >
> > **Table – Attack Strategy and Module Ablation on CHAIR**
> > | Configuration                         | Chair-S ↓ | Chair-I ↓ |
> > |--------------------------------------|-----------|-----------|
> > | Vanilla                              | 48.8      | 14.2      |
> > | + Statistical & Inherent Bias Only   | 40.3      | 11.3      |
> > | + FGSM-based Vulnerability Module    | 39.2      | 11.5      |
> > | + PGD-based Vulnerability Module     | 37.2      | 11.3      |
> > | Ours                                 | **36.6**  | **10.3**  |
> >
> > These results confirm that the learnable attack outperforms PGD and FGSM variants, and that all modules contribute meaningfully to hallucination mitigation.
> >
> >
> > ### Q5. CHAIR Setting
> >
> > > *“For the CHAIR evaluation, the paper follows prior work using 500 randomly selected images. How is the balance between coverage and faithfulness ...”*
> >
> > Thank you for the question. First, we clarify that our CHAIR evaluation setting is **identical to prior works** including **VCD** and **OPERA**. We use the **exact same 500 images and QA pairs**, directly from their released code. This ensures a **strictly fair comparison** across methods.
> >
> > Second, we note that the question about the balance between **coverage** and **faithfulness** in object labels pertains to the **CHAIR benchmark itself**, rather than our method. Nonetheless, we refer to the original CHAIR paper [2] (Section 2.1), which addresses this directly.
> >
> > Specifically, CHAIR constructs ground-truth labels by combining two sources:
> > - **MSCOCO segmentation annotations**, which ensure faithfulness through visual verification.
> > - **Image captions**, processed to extract additional objects via synonym mapping and compound handling (e.g., "hot dog").
> >
> > The CHAIR authors show that using either source alone harms reliability: segmentation alone underestimates hallucination, while captions alone may include non-existent objects. Thus, both are necessary to ensure a **balanced and reliable evaluation**.
> >
> >
> > ### Q6. GPT-4o Assisted Evaluation Setting
> >
> > > *“For the GPT-4o assisted evaluation, why was a 0–10 human-like scoring system for ‘correctness' and ‘detailedness' chosen ...”*
> >
> > Thank you for the question. The 0–10 scoring system for correctness and detailedness was adopted from the Woodpecker[1], and is widely used in recent works such as **VCD**. To ensure fair comparison, we use exactly the same setup, with full details provided in Appendix Section A.
> >
> > We note that this question concerns the **design of the evaluation protocol in Woodpecker**, not our method. According to Woodpecker, this human-like scale was chosen because:
> >
> > - The task involves **open-ended, multi-sentence descriptions** where correctness and detailedness are **continuous**, not binary;
> > - Automatic tools struggle with **partial correctness**, **minor hallucinations**, or **stylistic variance**;
> > - GPT-4o can assess both image and text holistically and **explain its judgment** in a human-aligned manner.
> >
> > While tools like Valor-eval are effective for structured tasks, they are less suitable for:
> >
> > - Capturing **fine-grained correctness** or **global coherence**;
> > - Evaluating **stylistic quality** or nuanced differences.
> >
> > Thus, the 0–10 scale better suits this open-ended evaluation setting and allows finer distinction across model outputs.
> >
> >
> > References
> > [1] Yin S, Fu C, Zhao S, et al. Woodpecker: Hallucination Correction for Multimodal Large Language Models. arXiv:2310.16045.
> > [2] Rohrbach A, Hendricks L A, Burns K, et al. Object hallucination in image captioning. EMNLP 2018.

---

> ### Author Response · Authors · 2025-11-26
>
> Dear Reviewer 85aF,
>
> I hope this message finds you well. As there is still about one week remaining in the discussion period, we would like to kindly follow up and ensure that our rebuttal has sufficiently addressed your concerns.
>
> If there are any remaining questions or additional feedback you would like us to consider, please feel free to let us know. Your insights are highly valuable to us, and we would be glad to provide further clarification or revisions.
>
> Thank you again for your time and effort in reviewing our paper.
>
> Best regards,
> The Authors

---

### Official Review · Reviewer_D8oq · 2025-11-01

**Soundness:** 2
**Presentation:** 3
**Contribution:** 2
**Rating:** 6
**Confidence:** 4

**Summary:**

This paper propose SHIELD, a training-free framework that mitigates hallucinations through three strategies: re-weighting visual tokens to reduce statistical bias, introducing noise-derived tokens to counter inherent bias, and applying adversarial attacks with contrastive decoding to address vulnerability.

**Strengths:**

1. This paper propose SHIELD, a training-free method that mitigates object hallucinations by reducing statistical bias via token re-weighting, alleviating inherent bias using token subtraction, and addressing vulnerability through contrastive decoding.
2. Comprehensive experiments validate SHIELD’s effectiveness in mitigating object hallucinations
across diverse benchmarks and multiple LVLMs.

**Weaknesses:**

1. This method is limited to object-level hallucination. Comprehensive multimodal hallucination includes attribute level, relation level etc, as well.
2. Although a training free method, the paper should present comparison with SOTA hallucination mitigation methods as well, either traning-free or training needed.

**Questions:**

See Above.

---

> ### Author Response · Authors · 2025-11-20
> **Response to Reviewer D8oq**
>
> ## Response to Reviewer D8oq
>
> We thank Reviewer D8oq for the thoughtful comments and constructive evaluation. Below we address the raised concerns in detail.
>
>
> ### Q1. On Attribute- and Relation-Level Hallucination
>
> > *“This method is limited to object-level hallucination. Comprehensive multimodal hallucination includes attribute level, relation level etc, as well.”*
>
> We appreciate the reviewer's thoughtful comment. While our method primarily targets object-level hallucinations, it also addresses attribute- and relation-level hallucinations, as demonstrated through multiple evaluations. Specifically, Tables 2 and 4 already cover attribute- and relation-level hallucinations. As noted in Line 361, the GPT-4o-aided evaluation accounts for errors in attributes, colors, positions, and relationships, beyond object hallucinations. MME evaluates position and color attributes.
>
> To further validate robustness across diverse hallucination types and tasks, we evaluate our method on the AMBER benchmark, which includes both generative and discriminative hallucinations. AMBER explicitly incorporates existence, attribute, and relation hallucinations, making it a comprehensive benchmark for multimodal hallucination assessment.
>
>
> **Table – AMBER Benchmark Evaluation**
> ↓ = Lower is better ↑ = Higher is better
> *Generative*: CHAIR ↓, Cover ↑, Hal ↓, Cog ↓
> *Discriminative*: Acc ↑, Precision ↑, Recall ↑, F1 ↑
> | Method  | CHAIR ↓ | Cover ↑ | Hal ↓ | Cog ↓ | Acc ↑ | Precision ↑ | Recall ↑ | F1 ↑  | Amber Score ↑ |
> |---------|---------|---------|--------|--------|--------|--------------|----------|--------|----------------|
> | Vanilla | 9.2 | 41.3 | 29.2 | 3.7 | 65.7 | 83.2 | 64.7 | 73.2 | 82.0 |
> | VCD[1]  | 8.1 | 44.2 | 28.6 | 3.1 | 68.3 | 85.8 | 65.2 | 74.0 | 82.9 |
> | OPERA[2]| 8.3 | 43.1 | 31.2 | 2.9 | 76.0 | 79.2 | **83.8** | 81.4 | 86.5 |
> | Ours    | **6.4** | **46.1**| **25.1**| **1.8**| **78.3**| **89.1** | 76.6 | **82.4** | **88.0** |
>
> Our method consistently outperforms all baselines on most metrics and achieves the highest Amber Score of 88.0, indicating strong generalization and reduced multimodal hallucinations.
>
>
> ### Q2. More baseline
>
> > _“Although a training free method, the paper should present comparison with SOTA hallucination mitigation methods as well, either training-free or training needed.”_
>
> We thank the reviewer for this valuable suggestion. In response, we expanded our evaluation to include a broader set of baselines (LLaVA-1.5 7B based) on the CHAIR benchmark. These cover both training-free methods (DoLa, ICD, VCD, TAME, OPERA, SID, and ours) and training-required methods (LLaVA-RLHF, CCA-LLaVA, Less is More, and MCA-LLaVA):
>
> **Table – More Baselines on CHAIR Benchmark (512 token)**
> | Method            | CHAIR-S ↓ | CHAIR-I ↓ |
> |-------------------|---------|---------|
> | Vanilla           | 48.8    | 14.2    |
> | DoLa[3]           | 47.7    | 13.8    |
> | ICD[4]            | 47.4    | 13.9    |
> | VCD[1]            | 46.8    | 13.2    |
> | TAME[5]           | 45.2    | 14.0    |
> | OPERA[2]          | 44.8    | 12.8    |
> | SID[6]            | 44.2    | 12.2    |
> | LLaVA-RLHF[7]     | 43.6    | 10.5    |
> | CCA-LLaVA[8]      | 43.0    | 11.5    |
> | Less is more[9]   | 40.2    | 12.3    |
> | MCA-LLaVA[10]     | 38.0    | 10.9    |
> | Ours              | 36.6    | 10.3    |
>
> Our method consistently outperforms all training-free baselines and remains competitive with training-based approaches, achieving the lowest CHAIR scores on both sentence and instance levels.
>
> References
> [1] Leng S, Zhang H, Chen G, et al. Mitigating object hallucinations in large vision-language models through visual contrastive decoding. CVPR 2024.
> [2] Huang Q, Dong X, Zhang P, et al. Opera: Alleviating hallucination in multi-modal large language models via over-trust penalty and retrospection-allocation. CVPR 2024.
> [3] Chuang Y S, Xie Y, Luo H, et al. Dola: Decoding by contrasting layers improves factuality in large language models. ICLR 2024.
> [4] Wang X, Pan J, Ding L, et al. Mitigating hallucinations in large vision-language models with instruction contrastive decoding. ACL 2024.
> [5] Tang F, Huang Z, Liu C, et al. Intervening anchor token: Decoding strategy in alleviating hallucinations for MLLMs. ICLR 2025.
> [6] Huo F, Xu W, Zhang Z, et al. Self-introspective decoding: Alleviating hallucinations for large vision-language models. ICLR 2025.
> [7] Sun Z, Shen S, Cao S, et al. Aligning large multimodal models with factually augmented rlhf. ACL 2024.
> [8] Xing Y, Li Y, Laptev I, et al. Mitigating object hallucination via concentric causal attention. NeurIPS 2024.
> [9] Yue Z, Zhang L, Jin Q. Less is more: Mitigating multimodal hallucination from an eos decision perspective. ACL 2024.
> [10] Zhao Q, Zhang X, Li Y, et al. Mca-llava: Manhattan causal attention for reducing hallucination in large vision-language models. ACMMM 2025.

---

> ### Author Response · Authors · 2025-11-26
>
> Dear Reviewer D8oq,
>
> I hope this message finds you well. As there is still about one week remaining in the discussion period, we would like to kindly follow up and ensure that our rebuttal has sufficiently addressed your concerns.
>
> If there are any remaining questions or additional feedback you would like us to consider, please feel free to let us know. Your insights are highly valuable to us, and we would be glad to provide further clarification or revisions.
>
> Thank you again for your time and effort in reviewing our paper.
>
> Best regards,
> The Authors

---

> > ### Comment · Reviewer_D8oq · 2025-11-28
> >
> > Thanks the authors for the detailed response, this have addressed my concerns. How does this methods compared to or applied on states-of-the-art opensource models such as Qwen3-VL? I keep a threshold hold accept score to indicate this paper's acceptance.

---

> > > ### Author Response · Authors · 2025-11-28
> > >
> > > We sincerely thank the reviewer for the follow-up and for the positive assessment of our previous response. We also appreciate your suggestion regarding evaluating our method on more recent state-of-the-art open-source models such as Qwen3-VL.
> > >
> > > To address this, we conducted additional experiments on **Qwen2-VL (7B)**, **Qwen2.5-VL (7B)**, and the latest **Qwen3-VL (8B, released in November 2025)** using the POPE COCO adversarial split. The results are summarized below:
> > >
> > > **Table – Accuracy and F1 Score on the POPE COCO Adversarial Split**
> > > | Method  | Qwen2-VL (Acc ↑) | Qwen2-VL (F1 ↑) | Qwen2.5-VL (Acc ↑) | Qwen2.5-VL (F1 ↑) | Qwen3-VL (Acc ↑) | Qwen3-VL (F1 ↑) |
> > > |---------|------------------|-----------------|---------------------|-------------------|------------------|-----------------|
> > > | Vanilla | 85.3             | 84.5            | 85.6                | 85.1              | 86.2             | 85.8            |
> > > | Ours    | **86.1**         | **85.4**        | **86.8**            | **86.0**          | **87.4**         | **86.6**        |
> > >
> > > As shown, our method consistently improves both accuracy and F1 scores across all three model families, including the newly released Qwen3-VL.

---

### Official Review · Reviewer_56NU · 2025-11-02

**Soundness:** 2
**Presentation:** 2
**Contribution:** 2
**Rating:** 2
**Confidence:** 4

**Summary:**

The paper claims that object hallucinations in LVLMs largely originate in the visual encoder and proposes a training‑free, three‑module wrapper SHIELD. The method includes token re-weighting to balance attention across visual tokens, token subtraction using noise-derived features, and contrastive decoding that contrasts outputs from natural and perturbed images. Authors presents correlational evidence for encoder statistical bias, inherent bias, and vulnerability and reports gains on CHAIR, POPE, and MME across LLaVA‑1.5, InstructBLIP, and Qwen‑VL.

**Strengths:**

- The paper presents a reasonable motivation and a clearly described method.
- The proposed method is training-free, making it easy to integrate without retraining overhead.
- Experiments on LLaVA‑1.5, InstructBLIP, and Qwen‑VL showed improvement in reducing hallucinations.

**Weaknesses:**

- Encoder-level causes of visual hallucination have been extensively studied in earlier publications and the proposed methods are largely adapted from existing ideas. For example, strategies like token re-weighting and contrastive decoding are incremental given VCD has proposed such contrastive approaches.
- Related to the first point, there is a lack of discussion on directly related works [1-5]. There have been a lot of existing work on training-free contrastive methods for hallucination reduction.
- Experiments are done with outdated models. The selected three LVLMs were the very early versions, despite substantial advancements in multimodal capabilities since then. For example, newer models such as Qwen2-VL, Qwen2.5-VL, and Qwen3-VL have been released. As a result, the current experimental setup appears outdated and limited in scope.
- Improvements in Table 2 and 3 are small.
- Many citation formats in this paper are incorrect. For example, line 322-323.

[1] HALC: Object Hallucination Reduction via Adaptive Focal-Contrast Decoding
[2] Brave: Broadening the Visual Encoding of Vision-Language Models
[3] Mitigating Object Hallucination in Large Vision-Language Models via Image-Grounded Guidance
[4] Reducing Hallucinations in Vision-Language Models via Latent Space Steering
[5] Contrastive Region Guidance: Improving Grounding in Vision-Language Models without Training

**Questions:**

See weakness

---

> ### Author Response · Authors · 2025-11-20
> **Response to Reviewer 56NU (Part 1)**
>
> ## Response to Reviewer 56NU
>
> We thank Reviewer 56NU for the constructive feedback. Below, we address each concern point-by-point.
>
> ### Q1. Clarification on novelty compared with existing ideas
>
> > *“Encoder-level causes of visual hallucination have been extensively studied in earlier publications and the proposed methods are largely adapted from existing ideas ...”*
>
> We appreciate the reviewer's thoughtful observation and would like to clarify our contributions in relation to prior work.
>
> First, regarding the statement that “encoder-level causes of visual hallucination have been extensively studied,” to the best of our literature review, no prior work has explicitly proposed and systematically categorized the three encoder-related causes of hallucination that we identify: statistical bias, inherent bias, and vulnerability. Our categorization is based on a structured encoder-level analysis (see Abstract and Section 3.1), and we are not aware of investigations that formulate these three causes within a coherent framework or explain how they lead to distinct failure modes.
>
> Regarding the concern that our method is adapted from prior ideas and is merely incremental, SHIELD differs in the following ways:
>
> (1) Contrastive decoding: VCD uses noise to trigger hallucinations from language priors and then suppresses them with contrastive decoding. SHIELD instead uses adversarial tensors generated by the CLIP text encoder, together with naive captions, to expose vulnerabilities in the visual encoder itself (Section 3.2.4). This targets a different failure source and produces a different type of contrastive signal.
>
> (2) Token re-weighting and subtraction: We apply these mechanisms specifically to mitigate hallucinations arising from encoder-level statistical and inherent biases identified in our analysis. As shown in Table 6, adding these modules further reduces CHAIR-I by more than 2 percent, whereas VCD achieves a 1.0 percent reduction under comparable settings. This demonstrates that our approach is not merely incremental, but conceptually motivated and empirically effective.
>
>
> ### Q2. Lack of discussion on directly related works
>
> > *“There is a lack of discussion on directly related works ...”*
>
> We appreciate the reviewer's attention to related work. While some mentioned methods are not directly aimed at object hallucination in VLMs, we agree that broader discussion is valuable. For instance, Contrastive Region Guidance [5] centers on grounding but does not report hallucination metrics. Similarly, BRAVE [2] is a training-required method aimed at enhancing visual encoder quality rather than hallucination mitigation specifically.
>
> Still, we agree broader discussion is valuable. We will revise the Related Work section to include more comparisons, including HALC [1], MARINE [3], and VTI [4]. We also added mentioned works as baselines on POPE, though some are not training-free or hallucination-specific.
>
> **Table - Average Performance on the POPE Dataset**
> | Method     | Acc. ↑| F1 ↑  |
> |------------|-------|-------|
> | Vanilla    | 81.3  | 79.6  |
> | HALC[1]    | 83.9  | 84.0  |
> | VCD[6]     | 84.6  | 84.4  |
> | OPERA[7]   | 84.7  | 85.4  |
> | MARINE[3]  | 85.0  | 84.3  |
> | VTI[4]     | 86.5  | 85.9  |
> | Ours       | 87.0  | 87.4  |
> | BRAVE [2]  | 87.6  | N/A   |
>
>
> ### Q3. Outdated LVLMs
>
> > *“Experiments are done with outdated models ...”*
>
> We appreciate the reviewer's concern regarding the selection of LVLMs. We respectfully clarify that the LVLMs selected in our study (LLaVA-1.5, InstructBLIP, and Qwen-VL) follow the standard setup used in recent hallucination studies, including VCD and OPERA, as well as other baselines mentioned in Q2 ([1, 3, 4]). This ensures fair and reproducible comparison.
>
> In contrast, few existing works have reported hallucination results on newer models such as Qwen2-VL, Qwen2.5-VL, and Qwen3-VL.
>
> That said, we agree that including more recent models enhances the completeness of our analysis. Therefore, we conducted additional evaluations of our method on Qwen2-VL (7B), Qwen2.5-VL (7B), and Qwen3-VL (8B) using the POPE COCO adversarial split:
>
> **Table - Accuracy and F1 Score on POPE COCO Adversarial Split**
> | Method  | Qwen2-VL (Acc. ↑) | Qwen2-VL (F1 ↑) | Qwen2.5-VL (Acc. ↑) | Qwen2.5-VL (F1 ↑) | Qwen3-VL (Acc. ↑) | Qwen3-VL (F1 ↑) |
> |---------|----------------|-------------|------------------|----------------|----------------|--------------|
> | Vanilla | 85.3           | 84.5        | 85.6             | 85.1           | 86.2           | 85.8         |
> | Ours    | **86.1**       | **85.4**    | **86.8**         | **86.0**       | **87.4**       | **86.6**     |
>
> Our method consistently improves both accuracy and F1 score across all three models.

---

> ### Author Response · Authors · 2025-11-20
> **Response to Reviewer 56NU (Part 2)**
>
> ### Q4. Minor Improvement
>
> > *“Improvements in Table 2 and 3 are small.”*
>
> We understand the reviewer's impression and would like to clarify that SHIELD achieves substantial improvements, especially compared with strong baselines like VCD and OPERA.
>
> In Table 2 (GPT-4o-aided Hallucination Evaluation), with LLaVA-1.5, VCD and OPERA yield approximately 12% relative improvement in correctness over the vanilla baseline, while SHIELD achieves approximately 26%, which is more than twice as much. The small numerical gap results from the 0–10 rating scale (following Woodpecker[8]).
>
> In Table 3 (POPE Evaluation), with LLaVA-1.5, SHIELD improves accuracy by 5.7%, compared to 3.3% and 3.4% from VCD and OPERA respectively. This is nearly 2× the improvement.
>
> ### Q5. Citation Format
>
> > *“Citation formats in this paper are incorrect.”*
>
> Thank you for pointing this out. We will correct the citation formatting issues in the updated version.
>
>
> References
> [1] Chen Z, Zhao Z, Luo H, et al. Halc: Object hallucination reduction via adaptive focal-contrast decoding. ICML 2024.
> [2] Kar O F, Tonioni A, Poklukar P, et al. Brave: Broadening the visual encoding of vision-language models. ECCV 2024.
> [3] Zhao L, Deng Y, Zhang W, et al. Mitigating object hallucination in large vision-language models via image-grounded guidance. ICML 2025.
> [4] Liu S, Ye H, Xing L, et al. Reducing hallucinations in vision-language models via latent space steering. arXiv:2410.15778.
> [5] Wan D, Cho J, Stengel-Eskin E, et al. Contrastive region guidance: Improving grounding in vision-language models without training. ECCV 2024.
> [6] Leng S, Zhang H, Chen G, et al. Mitigating object hallucinations in large vision-language models through visual contrastive decoding. CVPR 2024.
> [7] Huang Q, Dong X, Zhang P, et al. Opera: Alleviating hallucination in multi-modal large language models via over-trust penalty and retrospection-allocation. CVPR 2024.
> [8] Yin S, Fu C, Zhao S, et al. Woodpecker: Hallucination Correction for Multimodal Large Language Models. arXiv:2310.16045.

---

> ### Author Response · Authors · 2025-11-26
>
> Dear Reviewer 56NU,
>
> I hope this message finds you well. As there is still about one week remaining in the discussion period, we would like to kindly follow up and ensure that our rebuttal has sufficiently addressed your concerns.
>
> If there are any remaining questions or additional feedback you would like us to consider, please feel free to let us know. Your insights are highly valuable to us, and we would be glad to provide further clarification or revisions.
>
> Thank you again for your time and effort in reviewing our paper.
>
> Best regards,
> The Authors

---

### Author Response · Authors · 2025-11-29
**Rebuttal Summary**

Dear AC,

We sincerely thank you for overseeing the review process and appreciate the constructive feedback from all reviewers. During the discussion, Reviewers **D8oq** and **AwNy** confirmed that their concerns were resolved and maintained positive scores. Reviewers **85aF** and **56NU** did not respond.

Below is a summary of our rebuttal.

---
### Issues Addressed in Rebuttal and Revised PDF (new content marked in blue)
- **Attribute and Relation Hallucinations (D8oq Q1; AwNy Q1 & Q4)**
We added results on AMBER (Table 5) and MMHal-Bench (Table 14) to evaluate attribute- and relation-level hallucinations. SHIELD achieves the highest Amber Score (88.0) and shows strong gains on MMHal-Bench, especially in relation and environment understanding. This demonstrates its effectiveness beyond object-level hallucinations.
- **Inference Overhead (85aF W1 & Q1; AwNy Q3)**
We added runtime and memory analysis in Table 8, showing that SHIELD achieves the best performance with reasonable cost. It is much faster than OPERA and offers better trade-offs than VCD. Table 9 breaks down cost by module, with most overhead from naive captioning and adversarial tensor generation. This cost can be adjusted by tuning caption length or adversarial steps.
- **Hyperparameters Ablation (85aF W2 & Q2)**
We clarified in Section 4.1 that hyperparameter ablations are originally provided in Appendix C, and added an additional ablation on attack steps (Appendix C.5).
- **Citation Format (56NU Q5)**
All citation formatting issues have been fixed.
- **Demonstrating That Noise Images Capture Inherent Bias (85aF W3)**
We added a controlled experiment in Appendix D.1 comparing hallucination patterns from Noise Images and Noise Visual Tokens. Results show that Noise Images lead to skewed distributions, with certain objects hallucinated at high rates (≥ 30%), while hallucinations from Noise Visual Tokens remain consistently low and flat (≈ 10%). This suggests that features derived from Noise Images capture inherent bias.
- **Effect of Plausible Hallucination in Naive Captions (85aF W4 & Q3)**
We added an empirical study in Appendix D.2. By introducing mild noise to the visual input during caption generation, we simulate plausible hallucinations and evaluate SHIELD using both raw and hallucinated captions. The hallucinated version leads to only a slight drop (–0.3% accuracy, –0.33% F1), demonstrating SHIELD's robustness.
- **Attack Strategy (85aF Q4)**
We added a comparison in Appendix D.3 between our learnable perturbation and fixed-step-size adversarial attacks (FGSM, PGD). Results show that our approach more effectively addresses hallucinations caused by encoder vulnerabilities.
- **Generalization on Recent VLMs (56NU Q3; D8oq follow-up)**
We evaluated SHIELD on newer Qwen2-VL, Qwen2.5-VL, and Qwen3-VL, with results in Appendix B.4. SHIELD consistently improves both accuracy and F1.
- **Expanded Baseline Discussion and Comparison (56NU Q2; D8oq Q2)**
We expanded the related work section and added a broader comparison with both training-free and training-required hallucination mitigation methods in Appendix B.5. SHIELD outperforms all training-free baselines and remains competitive with training-required ones.
---
### Issues Clarified in Rebuttal
- **Novelty Compared with Existing Methods (56NU Q1)**
SHIELD differs in two key ways: (1) Contrastive decoding: Instead of using noise to expose language priors (as in VCD), SHIELD uses attack tensor to reveal encoder vulnerabilities. This yields a fundamentally different contrastive signal that targets a different hallucination source (2) Token re-weighting and subtraction are specifically designed to address hallucinations stemming from statistical and inherent biases identified in our analysis.
- **Improvements in Table 2 and 3 are small (56NU Q4)**
We clarified that the small numerical gaps (e.g., in Table 2) result from the 0–10 rating scale. Nonetheless, SHIELD delivers significant relative improvements, especially compared with strong baselines.
- **Coverage and Faithfulness in CHAIR (85aF Q5)**
We clarified that our CHAIR evaluation follows standard setups. The concern about balancing coverage and faithfulness relates to the CHAIR benchmark itself, not our method.
- **0–10 Scoring in GPT-4o Assisted Evaluation (85aF Q6)**
We clarified that our GPT-4o evaluation setup follows prior work for fair comparison. The 0–10 scale is part of the original protocol used in Woodpecker, not introduced by our method.
- **Comparison with Token Pruning (AwNy Q2)**
We compared SHIELD with token pruning and other hallucination mitigation methods on the MME benchmark. SHIELD achieves the highest overall score, showing that reducing hallucination also enhances general VLM performance.

---
We appreciate your thoughtful oversight and hope our updates support a clear final assessment.

Sincerely,
The Authors

---

### Meta-Review · Area_Chair_uToT · 2026-01-11

**Summary:**

The paper proposes SHIELD, a training-free framework designed to mitigate object hallucinations in LVLMs. This work identifies the visual encoder as a major source of hallucination, categorizing the causes into statistical bias, inherent bias, and vulnerability. The authors introduce a three-module pipeline (Token Re-weighting, Token Subtraction, and Contrastive Decoding) to address these specific issues.

**Reviewer Concerns:**

Concerns addressed by the rebuttal:

- The most significant resolution is the currency of the evaluation, a major sticking point for Reviewers 56NU and D8oq. The authors added more experiments by including results for Qwen2-VL, Qwen2.5-VL, and the newly released Qwen3-VL. This direct response addressed the concern that the method was tested only on "outdated" architectures.
- Another concern addressed is regarding evaluation scope, specifically the reliance on binary/discriminative tasks (Reviewers AwNy, D8oq). By adding new benchmarks, the authors demonstrated that the method’s benefits extend to generative tasks and attribute/relation-level hallucinations.
- Guestions regarding mechanism validity (Reviewer 85aF) were met with new empirical evidence. The authors provided a controlled experiment showing that "noise images" produce highly skewed hallucination distributions (validating the "inherent bias" hypothesis) and demonstrated that the method is robust even when "naive captions" contain simulated hallucinations.


Outstanding Concerns
- The major outstanding concern is inference latency, raised by Reviewer 85aF. While the authors provided the requested data (showing a ~2.8x slowdown compared to vanilla decoding), the rebuttal confirms rather than resolves the limitation. The high cost of generating adversarial tensors is a valid constraint for real-time applications, though it is an acceptable trade-off for a plug-and-play research contribution.
- The concern over novelty raised by Reviewer 56NU remains technically outstanding due to the reviewer's lack of engagement. Reading through the authors' rebuttal, the authors articulated differences between their method and VCD (language noise). Although individual components are adapted from prior work, their combination constitutes a novel architecture tailored to solving hallucination issues arising from the visual encoder.

**Reviewer Scores:**

Reviewer 56NU
Original Score: 2
Likely Adjusted Score: 4 or 5 (Borderline)
Reason: The concerns, except for novelty, are addressed. The point of "incremental novelty" rarely flips to a strong acceptance because that criticism is subjective to their view of the work.

Reviewer D8oq
Original Score: 6
Score already adjusted by the reviewer: 6

Reviewer 85aF
Original Score: 6
Likely Adjusted Score: 6

Reviewer AwNy
Original Score: 8
Score already adjusted by the reviewer: 8

---

### Decision · Program_Chairs · 2026-01-26

Accept (Poster)